# Young adult-born neurons improve odor coding by mitral cells

H. Shani-Narkiss [1], A. Vinograd[1,2], I. D. Landau [1], G. Tasaka[1], N. Yayon [1], S. Terletsky[2], M. Groysman[1], I. Maor[1], H. Sompolinsky[1,3] & A. Mizrahi [1,2 ✉]

New neurons are continuously generated in the adult brain through a process called adult neurogenesis. This form of plasticity has been correlated with numerous behavioral and cognitive phenomena, but it remains unclear if and how adult-born neurons (abNs) contribute to mature neural circuits. We established a highly specific and efficient experimental system to target abNs for causal manipulations. Using this system with chemogenetics and imaging, we found that abNs effectively sharpen mitral cells (MCs) tuning and improve their power to discriminate among odors. The effects on MCs responses peaked when abNs were young and decreased as they matured. To explain the mechanism of our observations, we simulated the olfactory bulb circuit by modelling the incorporation of abNs into the circuit. We show that higher excitability and broad input connectivity, two well-characterized features of young neurons, underlie their unique ability to boost circuit computation.

[1] The Edmond and Lily Safra Center for Brain Sciences, The Hebrew University of Jerusalem, Jerusalem, Israel. [2] Department of Neurobiology, The Hebrew University of Jerusalem, Jerusalem, Israel. [3] Racah Institute of Physics, The Hebrew University of Jerusalem, Jerusalem, Israel. ✉email: Mizrahi.adi@mail.huji.ac.il

New neurons are continuously generated in the dentate gyrus of the hippocampus and the subventricular zone (SVZ)[1–3]. In the olfactory system of rodents, neurons migrate from the SVZ and integrate into the olfactory bulb (OB)[4–6]. The vast majority of SVZ adult-born neurons are granule cells (abGCs), which develop to be fully functional inhibitory interneurons in the OB[7–10].

The role of abNs in olfaction has been implicated by virtue of their effects on numerous behavioral phenomena[11]. For example, short-term memory[12], olfactory perceptual learning[13], fine olfactory discrimination[7,14–16], odor-reward associations,[17] and innate behaviors[18,19] have all been shown to be affected after various manipulations of abNs. The breadth of these behavioral studies suggests that abNs have a strong impact on the general circuitry, and that abNs impact basic computations.

AbGCs, like resident GCs, are inhibitory cells connected through reciprocal dendro-dendritic synapses to mitral cells (MCs), the main output neurons of the OB[20–22]. In vitro studies established that abGCs inhibit MCs (e.g., Breton-Provencher et al.[12]; Mandairon et al.[23]). However, the contribution of this inhibition arising from the small sub-population to odor coding in vivo remains unclear. Two recent studies measured the contribution of newborn neurons to MCs physiological responses, but have shown heterogeneous results; optogenetic activation of abNs, achieved by virus infection at the SVZ, caused inhibition of the spontaneous activity of MCs but did not induce changes in their odor-evoked spiking responses[15]. Chronically depriving the whole process of adult neurogenesis using a genetic ablation approach showed a different effect on MCs. Neurogenesis-deprived mice showed a reduction of suppressive odor responses, consistent with their inhibitory nature, but only when animals were actively engaged in a behavioral task[14]. While these studies provide evidence for the direct involvement of abNs in MC function, a coherent interpretation of the different effects and a mechanistic explanation of the results remains an open problem to resolve.

We designed a new experimental system to genetically access and manipulate abGCs at different ages with high specificity and unprecedented efficiency. We reveal a surprising effect on MCs following transient silencing of abGCs. Specifically, silencing abGCs caused a seemingly paradoxical effect—sparsening and reduction in the magnitudes of both excited and suppressed odor-evoked responses by MCs, and a concomitant decrease in their discriminatory power. Effects were strong when young abGCs were silenced and decreased as they matured. Furthermore, we show that our results can be explained by a network model of adult neurogenesis in the OB, emphasizing the unique excitability and promiscuous input connectivity of young abGCs as critical factors. Thus, abGCs enhance MCs odor-evoked responses through a combination of their unique properties and the underlying network architecture of the OB.

## Results

**A new experimental system to label and manipulate adult-born neurons**. Common methods used to access and manipulate adult-born neurons (abNs) are often nonspecific and/or incomplete[24,25]. For example, direct infection of the SVZ/RMS with viruses has limited efficiency that is determined by the site of injection and is also prone to variability between experiments[26–29]. Genetic ablation methods have limited temporal resolution for causal manipulations and the underlying effects may partially result from compensation by the intact network[14,18,30,31].

In search for a robust method for targeting abNs efficiently and specifically, we tested several combinations of driver and reporter transgenic mice. One combination proved highly efficient—a

cross between the well-established Nestin-cre[ERT2] driver strain[32] with a recently published reporter strain, expressing tTA2 and histone− BFP following recombination (hereafter called "TB") (Tasaka et al.[33,34]). The Nestin-cre[ERT2] driver strain limits expression of Cre recombinase to abNs, and the TB strain provides tight regulation of cre-mediated recombination (see Tasaka et al.[34] for details). Recombination in neurons of TB mice is followed by expression of tTA2 and a nuclear blue fluorescent protein (histone2B-BFP-Myc), thus enabling tagging (either by BFP expression and/or by anti-Myc staining) and tTA2-based manipulation (Fig. 1a). A series of five tamoxifen injections administered to adult mice (180 mg per kg, once per day) induced high levels of recombination in the known neurogenic niches— the SVZ-OB pathway and the dentate gyrus of the hippocampus —with minor levels of labeling elsewhere in the brain (Fig. 1b, c). In the absence of tamoxifen, we found almost no Myc+ cells (Fig. 1c, d). The number of abNs in the OB increased steadily after tamoxifen injection (Fig. 1d, e). Since the number of recombined cells is also affected by the dose of tamoxifen, we measured the efficiency of labeling using the exact same levels of tamoxifen used in previous studies. Notably, in the Nestin-cre[ERT2] × TB line, the variability among mice was relatively low and absolute numbers of abNs were >>2-fold higher than the most robust and commonly used methods which are based on Nestin-cre[ERT2] mice that were described to date, and using the same tamoxifen administration protocol (Fig. 1e; ref. [32]). Thus, this system provides an experimental platform to access abNs with high efficiency and low noise, allowing us to manipulate a large and homogeneous population of abNs in vivo.

**Silencing abGCs decreases the number and magnitude of odor-evoked responses in MCs of anesthetized mice**. To study the role of abGCs on MC responses, we designed an experimental timeline to target abGCs at 4–8 weeks of age. In our system, tamoxifen acts like a switch to induce permanent expression of tTA2 in abNs, which then continuously accumulate in the OB. To limit the infected neurons to a specific age range, we used a tTA2-dependent virus injected directly to the OB at a specific time point. The age window of the neurons, which we study, is determined by the duration of time between tamoxifen injections (which is the birth date of the neurons), and the time of virus injection (which is the minimal age of the neurons being studied). Then, the time of imaging marks their age during the experiment (see timeline in Fig. 2a). For example, we infected new neurons with virus 4 weeks after tamoxifen administration (weeks post injection, WPI), when they were 0–4 weeks of age (Fig. 2a—Virus inj.) and imaged MCs, while silencing them, 4 weeks later when infected abGCs were 4–8 weeks of age (Fig. 2a—Imaging). Here, we chose to study new neurons at 4–8 weeks of age because during this developmental time window they are known to be fully integrated into the network[9,24,35,36], yet are still young and overly excitable, a property that fades away with maturation[24,35,37,38].

To image MCs while silencing young abGCs we combined chemogenetics and two-photon imaging. We injected the mitral and granule cell layers (GCLs) of the OB with two viruses, 4 weeks post tamoxifen injections with: (1) AAV1.TRE3G. hM4D (Gi)-mRuby to induce inhibitory DREADD expression in the 0–4 weeks old abGCs (Fig. 2b; abGCs are co-labeled with cytoplasmic mRuby and nuclear BFP), and (2) AAV5.CamKII. GCaMP6f, to induce expression of the calcium indicator GCamp6f in MCs (Fig. 2b; green cells). DREADD expression was highly specific to abGCs, averaging >95% infection specificity (Fig. 2c). The efficiency of silencing abGCs in this system was validated by assessing c-Fos expression following odor exposure

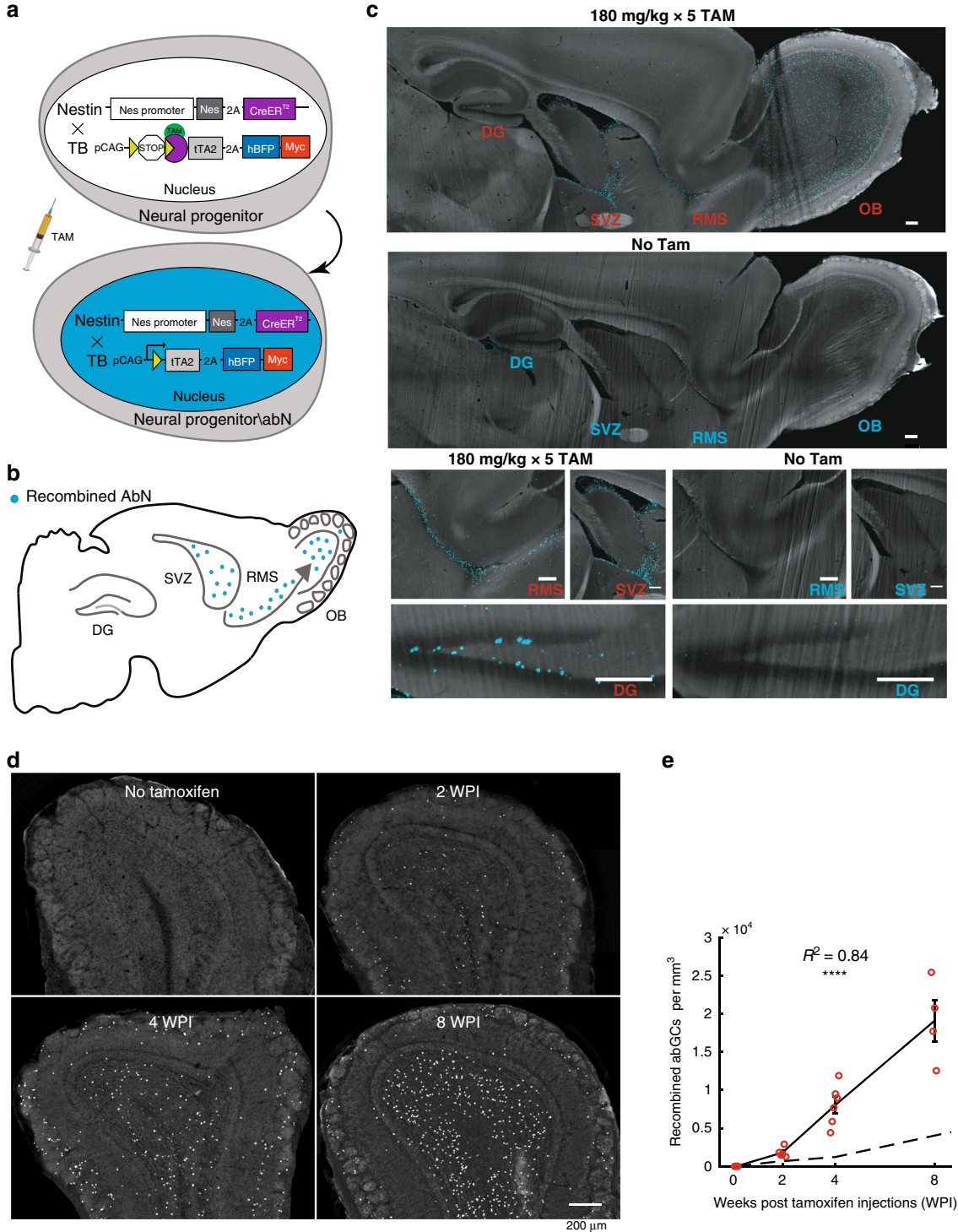

(Supplementary Fig. 1A–D) at different time points post virus injection. Specifically, CNO injection into Nest$^{Cre/+}$ x TB$^+$ mice that were exposed to odors significantly reduced the number and intensity of c-Fos in genetically tagged abGCs expressing DREADDs, as compared to abGCs that did not express DREADDs in the same mice ($N = 3$ mice; Supplementary Fig. 1C). Next, we evaluated MCs responses with and without the silencing of 4–8 weeks old abGCs. We tested both ketamine-anesthetized mice as well as awake mice (Figs. 2 and 3, respectively).

In anesthetized mice, we used Cre-negative siblings as controls (i.e., Nest$^{Cre/-}$ × TB$^+$, hereafter called NES−; $N = 10$ mice),

which received identical treatment to the experimental group (i.e., Nest$^{Cre/+}$ × TB$^+$, hereafter called NES+; $N = 10$ mice). We imaged MC responses to 11 odors (7 monomolecular odors, and 4 odor mixtures, five repetitions each), both before and 45 min after silencing abGCs via intraperitoneal injection of CNO (5 mg per kg; Fig. 2d–g). AbGCs, like their mature counterparts, make inhibitory synapses onto MCs[39]. Therefore, we expected to measure an excitatory effect when inhibiting abGCs (i.e., via disinhibition). Unexpectedly, inhibiting abGCs resulted, predominantly, in suppression of the calcium-evoked MC responses (Fig. 2e). On average, silencing abGCs decreased the number of odors each MC responded to, suggesting that abGCs broaden

**Fig. 1 A new experimental system to label and manipulate adult-born neurons. a** Schematic of the genetic components of the Nestin-creERT2 (Nes) driver mouse crossed to the tTA2-BFP (TB) reporter mouse. Tamoxifen (TAM) allows recombination in neural progenitors, which produce newly generated adult-born neurons (abNs). Cre-recombination will induce expression of tTA2 for an additional option of transgene expression (via TRE), and a nuclear blue fluorescent protein (BFP) tagged by Myc protein for detection in vitro. Top: Cre will be expressed in neural progenitors but will not induce recombination under normal conditions. Bottom: tamoxifen injection will induce expression of nuclear BFP and tTA2 in progenitors and abNs. **b** Schematics of the expected labeling in a Nestin+ × TB+ mouse after injection of tamoxifen. **c** Light-sheet micrographs (projection of 150 μm; using i-DISCO) of a sagittal view from Nestin+ × TB+ mice, following no TAM, and 4 weeks post TAM injections. AbNs nuclei are tagged by anti-Myc staining. Note that nonspecific staining is concentrated mainly around large blood vessels. All scale bars are 200 μm. **d** Confocal micrographs from the OB of Nestin+ × TB+ mice not injected with Tam, and at three different time points following Tam injections. Each dot is a single abN (nuclear BFP). **e** Quantification of abGCs in the OB after tamoxifen injections shows an addition of ~9000 abGCs per month. Red circles represent individual mice used for this experiment (counts from 18 mice in total, Pearson $R^2 = 0.84$, $p << 0.0001$). The dotted line is an approximation of the data described by Lagace et al.[32] using the same driver mouse crossed to a Rosa26-GFP reporter with the same tamoxifen administration protocol. SVZ sub ventricular zone, OB olfactory bulb, RMS rostral migratory stream, DG the dentate gyrus of the hippocampus. WPI weeks post tamoxifen injections. Error bars represent the standard error of the mean (SEM).

MCs tuning (Fig. 2h, $n = 321$ cells). In the temporal domain, abGCs silencing affected the peak response but not the latency to respond (Fig. 2i, $n = 1258$ cell-odor pairs). To test whether the reduction in MC responses was global or specific, we ranked odor responses by their magnitude for each MC individually. This analysis shows that silencing abGCs caused a global change in the responses of MCs (i.e., response magnitudes decreased similarly, regardless of their initial response magnitude; Fig. 2j). CNO injections to NES− mice induced only small and heterogeneous changes in individual MCs (Fig. 2g). The population averages were not significantly different before vs. after CNO injection (Fig. 2h, j—bottom traces, $n = 1558$ cell-odor pairs from 379 cells). The effects due to CNO injections were significantly different between the experimental and the control groups of mice (Fig. 2k, l) and consistent across individual mice (Supplementary Fig. 2B, C). Taken together, these results suggest an unexpected excitatory effect of abGCs on odor-evoked responses by MCs.

**Silencing abGCs quenches odor-evoked responses in MCs of awake mice.** Previous studies showed that odor-evoked responses by MCs and GCs are different in awake vs. anesthetized animals[40,41]. We, therefore, repeated the abovementioned experiments in awake mice. Specifically, we implanted mice (NES+, $n = 5$) with a chronic window and imaged MCs in a head-restrained, awake configuration (Fig. 3a). Since CNO by itself did not affect MC responses in awake mice that were not injected with DREADD (Supplementary Fig. 3), we used saline injections to the experimental mice as controls. In agreement with the results of the anesthetized state, silencing abGCs in awake mice induced suppression of odor-evoked calcium responses by the MCs (531 cell-odor pairs, $n = 170$ cells; $N = 5$ mice, Fig. 3b, d, e). Notably, in the awake state we also measured a significant proportion (≈40%; 404 cell-odor pairs; see "Methods" for definition) of suppressed calcium responses in MCs (e.g., Fig. 3b (top): cell-1 odor-1; cell-2 odor-1). Silencing abGCs also induced weakening of these suppressed responses (Fig. 3b, denoted by minus signs; Fig. 3d, e). Interestingly, the global effect, which we observed in anesthetized mice, remained symmetric, with respect to both suppressed (i.e., decreased activity following odor presentation) and excited responses (i.e., increased activity following odor presentation) (Fig. 3d, e). Thus, on average, MC responses are quenched (i.e., simultaneous decrease of excited and suppressed responses) when silencing the activity of abGCs (Fig. 3d, e). Mice injected with saline showed no significant change from baseline (Fig. 3b—saline; Fig. 3c–f—bottom panels). The effects due to CNO injections were significantly different than those of saline injections for the number of responses, and magnitudes of excited as well as suppressed responses (Fig. 3g, h).

**AbGCs improve odor discrimination by MCs.** We next asked if and how the changes we observed at the single cell level affected the information carried by MCs at the population level. To qualitatively evaluate the change in odor discrimination that is caused by suppression of abGCs, we calculated d′ (d prime) values among odor responses of the population and the corresponding CNO-induced changes. Figure 4 demonstrates (for graphical purpose only) a principle component analysis (PCA) for six arbitrary odors from a single mouse (Fig. 4a, b). To quantify the change in discriminability, we analyzed d′ values for all odor pairs from the $n$-dimensional space of the response, where $n$ is the number of neurons in each mouse (Fig. 4b, two left matrices; calculated from 52 neurons in this example mouse). Then, we calculated the difference between all d primes using a discrimination change index (DCI, Fig. 4b, right matrix). A decrease in discrimination efficiency is expressed as a negative DCI and vice versa. DCIs ≈ 0 refers to no change in discriminability among a pair of odors. The results of our analysis show that silencing abGCs caused a significant reduction in DCIs, that was evident for almost all pairs of stimuli (Fig. 4c, e; data from all mice). The same mice injected with saline showed heterogeneous differences, the average of which was in the opposite direction than the experimental group (Fig. 4d, e, see "Methods" referring to this point). In accordance with this observation, in anesthetized mice, silencing abGCs resulted in qualitatively similar but quantitatively weaker effects at the population level, and in control mice there were no significant differences at all (Supplementary Fig. 4A–C). These data are the first to provide causal evidence for a role of abGCs in promoting odor discrimination by MCs. These data, thus, provide a mechanistic insight to the multiple studies that assigned a necessary role for adult neurogenesis in behavioral discrimination tasks[7,15,16,42–45].

**The impact of abGCs on MCs odor coding gradually decreases with age.** One of the main open questions in the field of adult neurogenesis is concerned with its transient nature—how does the impact of newborn neurons on the network change with age? To answer this question, we exploited our experimental system further and silenced abGCs at different ages as they matured. To cover a range of ages around the developmental process, we compared three age groups—young (4–8 weeks old), established (8–12 weeks old), and mature (12–16 weeks old)[9,24,46]. Our system allowed us to evaluate the responses of the exact same MCs while silencing the same abGCs at different ages. Separate control experiments showed that DRAEDD efficiency was intact at both young and old ages (Supplementary Fig. 1A, D), and the numbers of DREAD-expressing neurons were not different in all ages (Supplementary Fig. 5D). We performed in vivo time lapse imaging, repeating the CNO-silencing experiment in awake

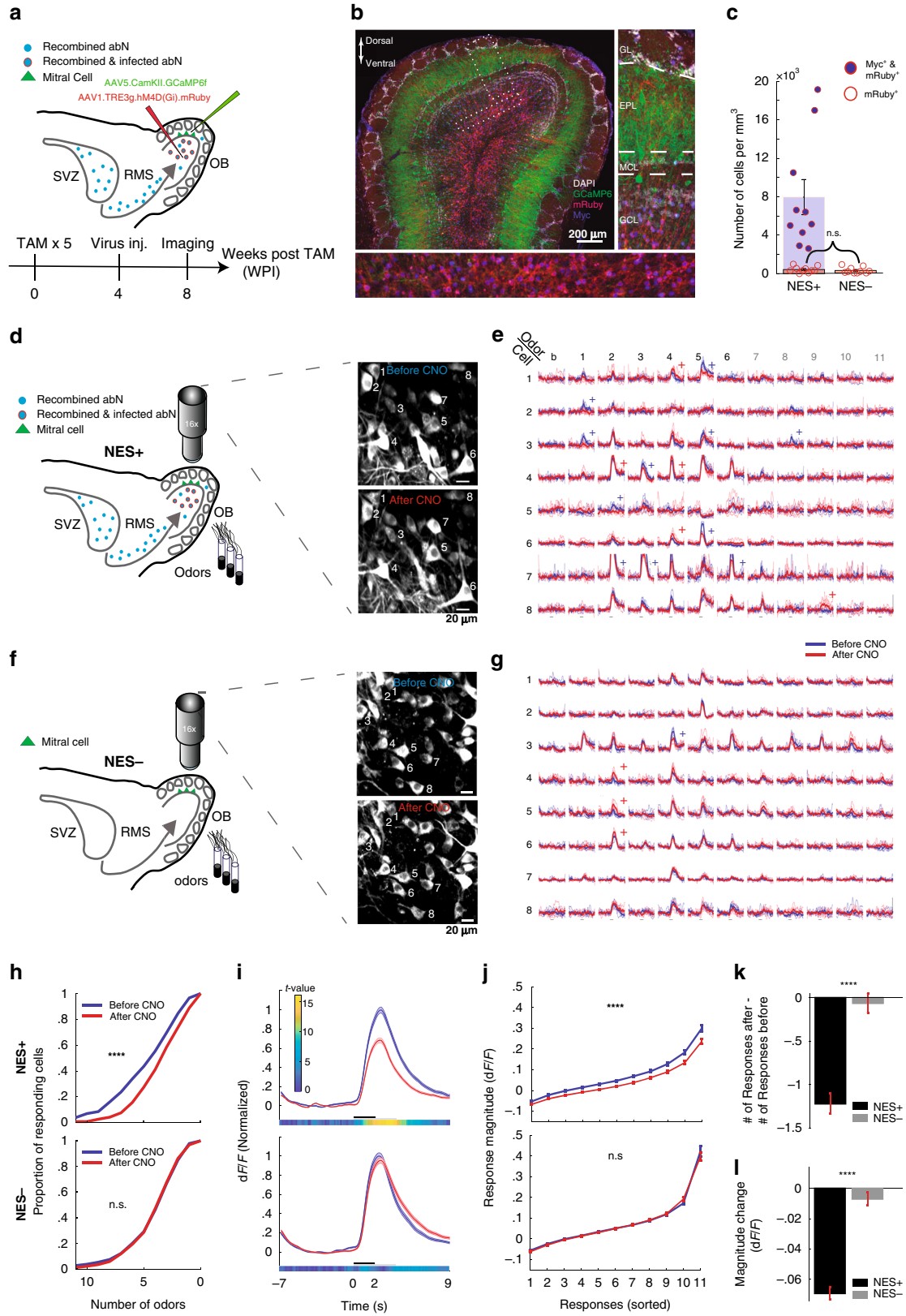

mice at 8, 12, and 16 WPI (Fig. 5a, b). Single MCs were tracked over time, having largely stable response profiles to the 11 odors (e.g., Fig. 5b; compare blue responses in 8, 12, and 16 WPI). The strong impact of silencing abGCs 4–8W of age, gradually decreased as they matured. The general effects of silencing abGCs were still evident at 8–12W of age, but waned by 12–16W of age

(Fig. 5c, d, f, g). The weakening of the effect by abGCs silencing was also evident as a decrease in the number of responses with significant changes in response magnitude following CNO injection (Fig. 5e). Similarly, the DCI values (calculated from the population d′) decreased at 12 WPI, and reached control levels by 16 WPI (Fig. 5h, compare with Fig. 4e). Thus, the age-dependent

**Fig. 2 Silencing abGCs decreases the number and magnitude of odor-evoked responses in MCs. a** Schematic illustration of the experiment and timeline. **b** Micrograph from the OB (4 WPI). Insets—magnification of the dotted-line boxes. PGL/EPL/MCL/GCL peri-glomerular, external plexiform, mitral and granule cell layers, respectively. **c** Quantification of DREADD-virus specificity (filled bars; $N = 20$ mice, $p = 0.17$, Mann–Whitney $U$ test). **d** A representative field of MCs expressing GCamp6f before and after CNO. **e** Examples of odor-evoked calcium transients before (blue) and after (red) CNO injection, from 8 neurons (shown in **d**) in response to 11 odors. Odor stimulation—2 s. Thin traces are five single trials; thick traces are means. Blue/Red plus signs mark a statistically significant difference between conditions, as a decrease or increase, respectively. Scale –100% dF/F. b blank. Odors: 1—valeraldehyde, 2—methyl propionate, 3—ethyl acetate, 4—butyraldehyde, 5—ethyl tiglate, 6—propanal, 7—TMT, 8—female urine, 9—male urine, 10—peanut butter, 11—pups beddings. **f, g** Same as **d, e** but from NES− mouse. **h** Cumulative distribution of the proportion of MCs responding to 0–11 odors, before (blue) and after (red) CNO, in NES+ (top) and NES− (bottom) groups (NES+: $n = 321$ cells, $p < 0.0001$. NES−: $n = 379$ cells, $p = 0.99$, Kolmogorov–Smirnov tests). **i** Average ± SEM responses to odor stimulation before and after CNO administration. CNO induced a general decrease in response in NES+ mice (1258 cell-odor pairs) but not in NES− (1558 cell-odor pairs) mice. Black line—odor presentation, gray line—response window. Colored bar—$t$ value between the before-after distributions as a function of time. **j** Response magnitude in ranked order, before and after CNO injection. The tuning curve is diminished only for NES+ mice (NES+: $n = 321$ cells, $p < 0.0001$; NES−: $n = 379$ cells, $p = 0.6$, Wilcoxon signed rank tests on cells curves' standard deviations before vs. after CNO) **k** Difference in responsiveness following CNO ($n = 700$ cells, $p << 0.0001$, unpaired $t$ test). **l** Difference in responses magnitude following CNO ($n = 2816$ cell-odor pairs, $p << 0.0001$, unpaired $t$-test). NES+: $N = 10$ mice; NES−: $N = 10$ mice, for all comparisons described in this figure. Statistical tests are two-sided, and error bars are SEMs.

impact of abGCs on individual MCs is also evident at the population level.

**A suggested mechanism by which abGCs increase information in MCs.** The abovementioned results raise two main questions: (1) how can two seemingly opposite effects arise from exclusively inhibitory neurons (i.e., increased and decreased excited and suppressed responses, respectively)? and (2) how can a small fraction of neurons have a rather large net effect on the population? To answer these questions, we built a simplified mathematical model of the OB, building upon the basic connectivity between MCs and GCs/abGCs by using biologically relevant parameters (Fig. 6a, b). The model was designed as a recurrent neural network with lateral inhibition mediated by reciprocal synapses between GCs/abGCs and MCs (Fig. 6a, b, see also "Methods" for full definition). In accordance with the literature, the ratio between the number of MCs and GCs was set to 1:100[47–49]. According to the estimate of abGC accumulation in our system and previous estimations regarding the total density of GCs in the OB (Fig. 1e, see also[10,32,47,50,51]) abGCs constituted 2.5% of the total GC population (Fig. 6b). MCs received differential odor input originating from olfactory sensory neurons, plus baseline input (Fig. 6b—black input arrows to MCs), and were reciprocally connected to GCs and abGCs (Fig. 6b—blue and red arrows), in a distance-dependent manner (Fig. 6c; Egger and Urban[52]; Huang et al.[53]). To model abGCs, we set their excitability to be higher and their input connectivity to be more promiscuous compared to resident GCs, as suggested by previous experimental work (Fig. 6c, d; refs. [38,54,55]).

In the model, GCs mediate lateral inhibition, which sharpens MC odor-evoked responses relative to their input. AbGCs play a significant role despite their small numbers due to their increased efficacy—i.e., the product of their increased excitability and input promiscuity relative to mature GCs. Silencing abGCs in the model led to quenching of MC odor-evoked responses in a similar manner to that observed in our empirical data (Fig. 6e, compare with Fig. 3e). Since silencing fully mature abGCs did not impact MCs activity in our empirical experiments, and since fully mature abGCs are known to have similar properties to those of resident GCs[9,24], we also silenced 2.5% of the mature GCs in our model. In accordance with our empirical observation (Fig. 5), silencing 2.5% of mature GCs did not change the shape of the tuning curve of MCs (Fig. 6e, green curve vs. blue curve).

Notably, the strength of general GC-mediated lateral inhibition plays an important role in our model. Increasing the strength of general GC-mediated lateral inhibition, further sharpens the

tuning of MCs output relative to their input. Importantly, the stronger the lateral inhibition, the higher the impact of abGCs increased efficacy on MCs tuning (Fig. 6f). For a fixed strength of lateral inhibition, the model suggests that increasing either the proportion of abGCs or their efficacy would further sharpen MC tuning (Fig. 6g).

Finally, to estimate how abGCs contribute to odor discrimination by MCs in our model, we calculated the Chernoff distance between simulated odors[56]. The Chernoff distance ($D_c$) is a measure of the difference between two probability distributions and is directly related to the ability to correctly differentiate between sensory inputs[57]. We show analytically that the impact of silencing abGCs on $D_c$ is approximately equal to their impact on tuning curve sharpness (see "Methods" and Supplementary Methods).

**Adult neurogenesis is energetically useful for efficient coding.** We were now able to assess the extent to which adult neurogenesis is efficient. For this purpose, we analyzed the possible tradeoff between MCs and abGCs population sizes, asking how this tradeoff would influence the discriminative power obtained by the network. We first plotted how $D_c$ changes for networks containing different numbers of MCs and abGCs (Fig. 6h, red and blue gradients). Comparing to a biologically plausible reference point of 1000 MCs and 2.5% (2500) abGCs (Fig. 6g, red dot), we find that adding abGCs to the network would allow a decrease in the number of MCs necessary to maintain a given level of discriminability between odors (Fig. 6g, black solid line represents the iso-discrimination curve). The rapidly decreasing slope of the iso-discrimination curve shows that the addition of abGCs is particularly efficient in low numbers. For example, half the total potential benefit of abGCs is achieved when they are just 7.5% of the total GC population (Fig. 6g, arrow). This suggests that the estimated values of physiological turnover (~10%) in the OB, efficiently balances an energy-information tradeoff. Notably, the iso-discrimination curve is essentially independent of the parameters in the model (see Supplementary Methods for mathematical proof). Taken together, these data explain not only how just a few neurons can contribute to odor coding in the mature OB, but also show the possible energetic advantages of adult neurogenesis.

**Discussion**
We established a new experimental system to label and manipulate abNs. Inherent to the system is its ability to access a large number of abNs for further experimentation. Using viruses,

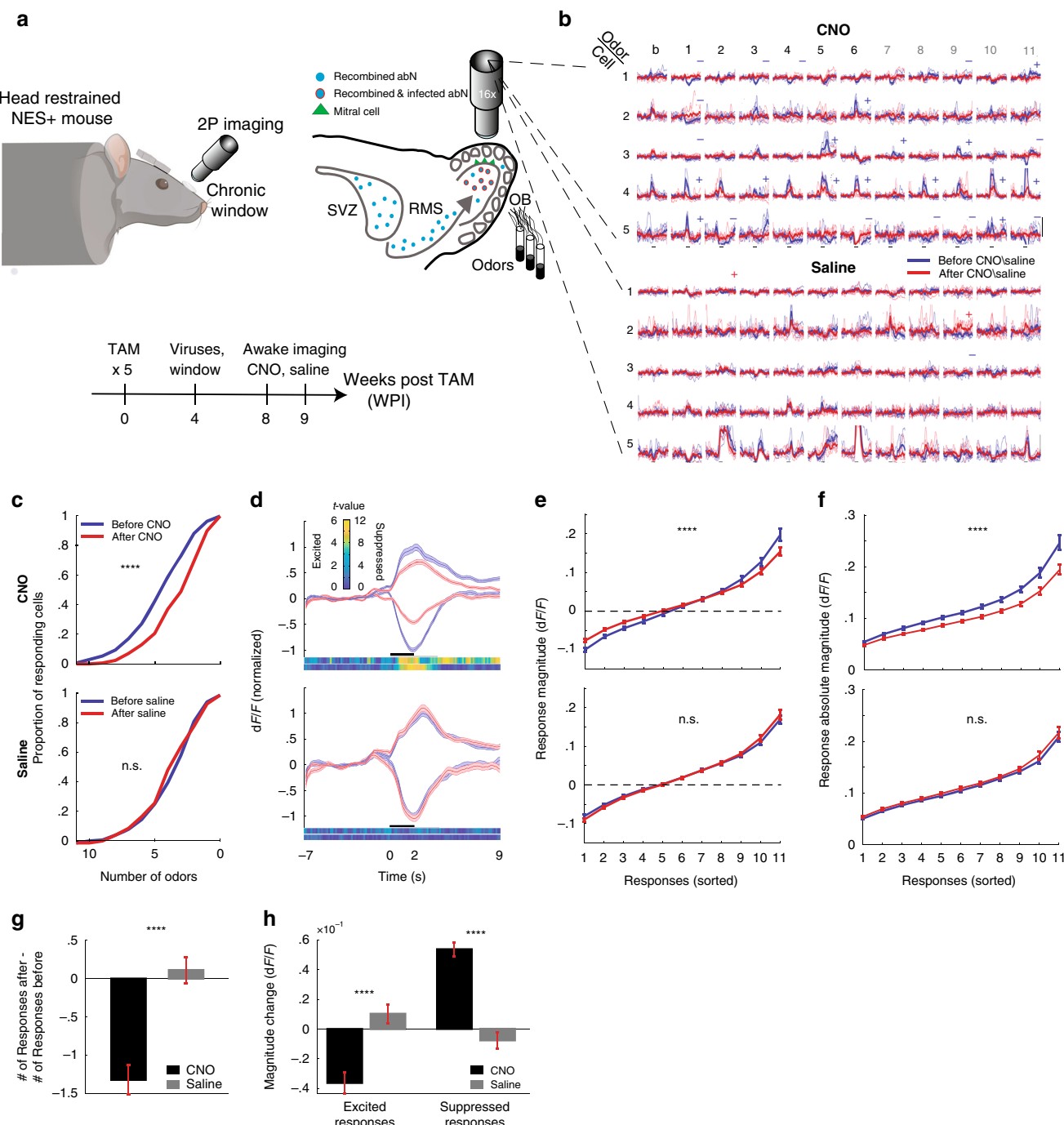

**Fig. 3 Silencing abGCs quenches odor-evoked responses in MCs (awake mice). a** Schematic illustration of the setup, experiment, and timeline.
**b** Examples of odor-evoked calcium transients for ten different neurons from the same mouse. All symbols are the same as in Fig. 2e, g, with the addition that minus signs denote significant differences among suppressed responses. **c** Cumulative distribution of the proportion of MCs responding to 0–11 odors, before (blue) and after (red) CNO (top) or saline (bottom) (CNO: $N = 5$ mice, $n = 170$ cells, $p < 0.0001$; Saline: $N = 5$ mice, $n = 152$ cells, $p = 0.62$, Kolmogorov–Smirnov tests). **d** Traces of the average ± SEM calcium responses to odor stimulation before and after CNO or saline administration. Suppressed (CNO: 404 cell-odor pairs, Saline: 251 cell-odor pairs) and excited (CNO: 531 cell-odor pairs, saline: 522 cell-odor pairs) responses are shown separately. Upper colored bar (excited responses, range: 0–6) and lower colored bar (suppressed responses, range: 0–12) are $t$ values between the before-after distributions as a function of time. **e** Response magnitude in ranked order for all cell-odor pairs showing quenching of both suppressed (negative values) and excited (positive values) responses. (CNO: $n = 170$ cells, $p \ll 0.0001$; saline: $n = 152$ cells, $p = 0.07$, Wilcoxon signed rank tests on cells curves' standard deviations before vs. after CNO). **f** Response magnitude in ranked order, in absolute values to account for both excited and suppressed responses equally. The values before injection are reduced after CNO but not after saline injection (CNO: $p \ll 0.0001$; saline: $p = 0.83$, Wilcoxon signed rank tests on cells curves' standard deviations before vs. after CNO). **g** Quantitative analysis of the difference in responsiveness (number of responses) due to CNO vs. saline. ($n =$ total 322 cells, $p \ll 0.0001$, unpaired $t$-test). **h** Quantitative analysis of the difference in response magnitude due to CNO vs. saline injections. Suppressed and excited responses are shown separately. $n_{excited} = 1053$ cell-odor pairs, $p \ll 0.0001$, $n_{suppressed} = 655$ cell-odor pairs, $p \ll 0.0001$, unpaired $t$-tests). CNO condition: $N = 5$ mice; saline condition: $N = 5$ mice, for all comparisons described in this figure. Statistical tests are two-sided, and error bars are SEMs. **a** was created with BioRender.com.

another layer of access can be achieved for higher spatio-temporal resolution and specificity (Figs. 1, 2, and Supplementary Fig. 1). As compared to other methods, this method is less variable and more efficient in the number of abNs being targeted[18,24,26,27]. Thus, as we show here for the OB, our system provides a sensitive, efficient, and reliable method to label and manipulate abNs. Since recombination is evident also in hippocampal abNs (Fig. 1c —bottom left), future studies could use this system to study neurogenesis in the dentate gyrus as well.

To date, the role of GCs in general and particularly abGCs in shaping MCs activity remains unclear[58–60]. One idea is that GCs predict the input from olfactory receptor neurons and establish a counterbalancing inhibitory activity that (incompletely) mirrors the activity of their cognate MCs inputs[61]. According to this theory, the odor-evoked activity by MCs is an error representation of the olfactory inputs that is modulated by GCs. In this scenario, excited and suppressed responses of MCs reflect two different error types. Positive errors reflect instances of incomplete inhibition by GCs, and negative errors occur when MCs receive excess inhibition by GCs. In the context of this theory, the symmetric nature of the effect caused by silencing abGCs in our data (Fig. 3e), assigns a role for abGCs as means to enhance both positive and negative error signals in the OB.

The activity and morphology of GCs are continuously shaped through their previous experience[23,45,62–64]. In fact, the most unique attribute of abGCs is their lack of experience. As such, they serve, mechanistically, as a continuous source of future synapses that shape the OB by experience. Specifically, when they are young and unexperienced, abGCs contribute to constant error signaling. Specifically, they boost MC responses regardless of the odor input encountered. This mechanism ensures that the network receives a continuous level of noise. Put differently, abGCs provide noise to the system and experience-dependent plasticity reduces this noise by strengthening the synapses of young abGCs. Under these conditions, and as long as abGCs continue to develop, odor representations are never completely stable.

Another, more parsimonious, idea for the role of abGCs is their ability to improve the way MCs discriminate among odors. As we observed empirically, abGCs increase the dynamic range of MCs, thereby improving discrimination. Notably, this is consistent with the effects observed in numerous causal behavioral experiments[14,15,45]. Better discriminatory power by MCs can serve downstream areas, improving multiple computations such as pattern separation[14,65].

How can the silencing of purely inhibitory neurons contributes to both decreased excited and suppressed odor-evoked responses? Our model suggests that while silencing abGCs increases stimulus-evoked firing $\left(r_i^{mc}(\text{odor})\right)$, it simultaneously causes an increase in spontaneous activity $\left(r_i^{mc}(0)\right)$. Thus, the effective contrast between stimulus-evoked firing and baseline firing becomes lower (Fig. 6e). More specifically, the quenching of the tuning curve observed in our model depends on the quantity used to evaluate MC responses: $R_i^{\text{MC}}(\text{odor}) = \frac{r_i^{\text{MC}}(\text{odor}) - r_i^{\text{MC}}(0)}{r_i^{\text{MC}}(0)}$. This definition, which is analogous to the imaging measurement (i.e., $df/f$), evaluates a response as a deviation relative to baseline, rather than an absolute firing rate. Thus, our model predicts that silencing abGCs, in addition to increasing odor-induced excitation of MCs, will also induce an increase in spontaneous firing by MCs. Remarkably, this inferred property was indeed observed empirically. This prediction is consistent with well-established findings from anatomy and slice electrophysiology, which showed that the abGCs-to-MCs synapses are purely inhibitory[12,15,23]. In addition, in vivo recordings of MCs spontaneous activity showed

a decrease in MCs activity when abNs were activated (Alonso et al.[15]). In our hands, using calcium fluorescence before odor presentation, we found that there is a significant, yet small, increase in basal fluorescence following CNO administration (Supplementary Fig. 2D), consistent with the electrophysiological studies.

In our model, while GCs provide lateral inhibition that sharpens the tuning of MCs (Fig. 6f), abGCs form a particularly strong pool of cells that provides a similar function to mature GCs but with higher efficacy. Importantly, as general GC-mediated inhibition is increased, the relative impact of abGCs becomes more dramatic (Fig. 6f, see how the color sensitivity along the $y$ axis increases for increasing values along the $x$ axis). This relationship may underlie the stronger effects we have measured when silencing abGCs in awake mice relative to anesthetized mice where GCs, as well as abGCs, are less active[40,54].

As abGCs mature, their excitability and promiscuity drop to mature GC levels[46,66]. This developmental process will result in a gradually diminishing impact these cells have on MC tuning (Fig. 6g, $x$ axis), consistent with our empirical results (Fig. 5). Our model also predicts a strong impact of increasing the proportions of abGCs on MC sharpening (Fig. 6g, $y$ axis). This result may explain, mechanistically, numerous biological phenomena in which increased/decreased levels of abNs were correlated with gain/loss of behavioral and cognitive performance, respectively (e.g., Ferreira et al.[67]; Leuner and Sabihi[68]; Mak and Weiss[19]; Van Praag et al.[69]; Moreno et al.[13]; Sakamoto et al.[18]; Shingo et al.[70]; Siopi et al.[71]).

Finally, the model suggests that continuous addition of young abGCs to the OB network allows efficient preservation of information in the network. In the presence of abGCs, fewer MCs are needed to preserve discriminatory power than would otherwise be necessary (Fig. 6h). This, therefore, is a novel insight to the yet mysterious role of adult neurogenesis in the brain. We argue that low levels of adult neurogenesis provides a large advantage for discrmination by the mature neural circuit.

Like abNs in the OB, young hippocampal neurons are also highly excitable. In both systems, low input specificity is gradually shaped toward highly specific connectivity[37,72,73]. Numerous functions were assigned to hippocampal abNs—from pattern separation (see review by Sahay et al.[65]) through contextual discrimination[74,75] up to forgetting[76]. Differences in the connectivity of the hippocampus and differences in the synaptic signature of abNs (in the hippocampus abNs are excitatory) will define the specific function of abNs in the hippocampus. However, our theoretical framework used to explain how a small proportion of young and excitable neurons contribute to efficient coding in mature circuits, may share common mechanisms with the hippocampus.

## Methods

**Animals**. All experimental procedures were approved by the Hebrew University Animal Care and Use Committee. Nestin-Cre$^{\text{ERT2}}$ (Jax stock #016261, background strain C57BL/6, Lagace et al.[32]) were crossed to TB mice (Jax stock #031776, background strain FVB; Tasaka et al.[34]). Breeding colonies were obtained by coupling heterozygous Nestin-Cre$^{\text{ERT2}}$ with homozygous TB breeders from both genders. Female offspring were genotyped for Cre and assigned to the different experiment conditions according to their genotype at 8 weeks old. Cre$^+$ mice were used for all experiments, except for the control group of the anesthetized experiment, in which Cre$^-$ mice were used. Genotyping was always verified by the presence or absence of BFP$^+$ cells, in vivo or in histological sections. Mice were 8 ± 1 weeks old in the beginning of the experiment. Temperature in animals housing was 21–24 °C, and 12/12 light–darkness regime was always maintained.

**Surgical procedures**. We anesthetized mice with an intraperitoneal injection of ketamine and medetomidine (100 and 0.83 mg per kg, respectively) and a subcutaneous injection of carprofen (4 mg per kg). In addition, we injected mice subcutaneously with saline to prevent dehydration. We assessed the depth of

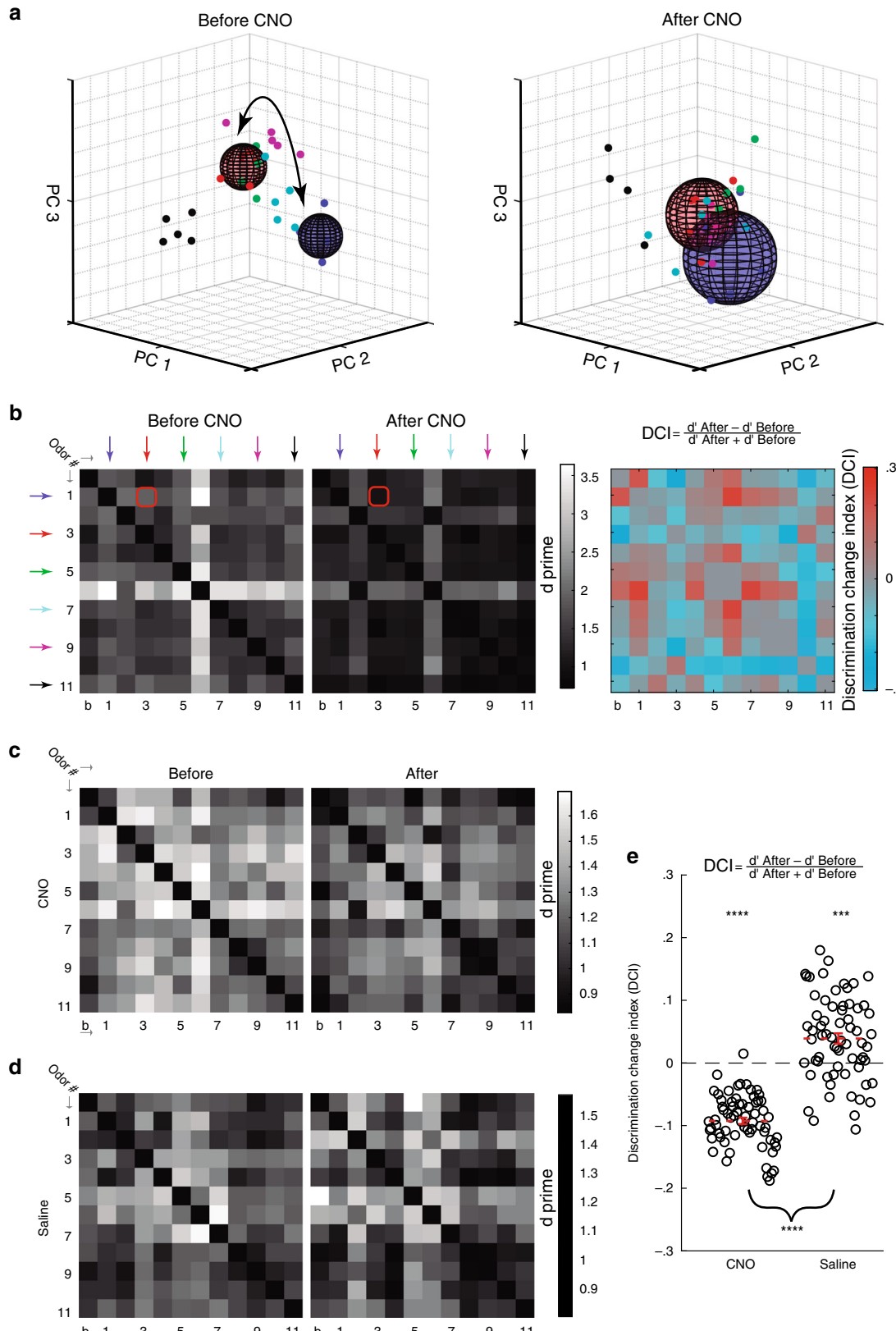

anesthesia by monitoring the pinch-withdrawal reflex and added ketamine/medetomidine to maintain it when needed. We continuously monitored the animal's rectal temperature at 36.5 ± 0.5 °C. For calcium imaging, we made a small incision in the animal's skin and glued a custom-made metal bar to the skull using dental cement to fix the head for imaging under the microscope. For acute imaging, we performed a craniotomy (2.5-mm diameter) over one OB. We poured 1.5% low-melting agar (type IIIa, Sigma-Aldrich) over the exposed brain, placed a glass cover over the craniotomy, and then secured it with dental cement. For awake experiments, we used triple layered window, following the procedure described by Goldey et al.[77].

**Tamoxifen administration**. Nestin-Cre$^{ERT2}$ × TB mice (8–9 weeks old) were administered with tamoxifen (TAM) at 180 mg per kg every day for 5 days

**Fig. 4 abGCs improve odor discrimination by MCs in awake mice. a** PCA representation from one mouse before (left) and after (right) CNO administration. For clarity, only six odors are shown in a 3-dimensional space of neural responses. Each dot is a trial; each color is an odor. Variance between different trials of the same odor (distance within) is represented as spheres, with a radius equal to the average distance of the dots from the centroid of their cluster. Separation between a certain pair of odors (distance between) is represented by a black arrow and is calculated as the Euclidean distance between the centroids of their clusters. The clusters of odors #1 (blue) vs. #3 (purple) are emphasized for the purpose of visualization only. **b** All matrices are calculated for the data shown in **a** only. Left 2 matrices showing all d-prime values in a pairwise manner before and after CNO. Colored arrows represent the odors shown in **a**. The red squares mark the example emphasized in **a**. Odor numbers as in Fig. 2b. Right—discrimination change index (DCI) matrix, in which each entry is calculated according to the presented DCI formula, from the data depicted in the two left matrices. **c** Matrices of d primes for all odor pairs, calculated in $n$-dimensional space ($n$ = number of cells) for each mouse and then averaged over all mice, before and after CNO. **d** Same as **c** for saline treatment. **e** DCI following either CNO or saline averaged over all mice for all possible pairwise comparisons. Both conditions were significantly different than 0 ($N$ = 66 DCIs for all comparisons, CNO vs. saline: $p \ll 0.0001$, CNO vs. 0: $p \ll 0.0001$, saline vs. 0: $p < .0001$, $t$-tests followed by Bonferroni correction). Statistical tests are two-sided, and error bars are SEMs.

(intraperitoneally; dissolved in 100% sunflower oil via ~1 h 200 rpm rotation in 37 °C accompanied by repetitive Vortex use).

**Light-sheet microscopy and i-Disco**. For whole brain imaging we used a Light-Sheet microscope (Ultra-microscope II, LaVisionBioTec) with fixed lens configuration using a ×4 lens. Images were acquired by an Andor Neo sCMOS camera (2560 × 2160, pixel size 6.5 × 6.5 μm, Andor) in 16 bit. Cleared brains were attached with epoxy glue to the sample holder and imaged at 10-μm steps along the Z axis. Autofluorescence was acquired with a 488-nm laser with 525/50 emission filter. The anti-Myc signal (Alexa-647) was acquired with a 640-nm laser with a 690/50 emission filter. Mice were perfused and brains collected for iDISCO clearing as described by others[78]. For Myc staining, we used a rabbit anti-Myc antibody (abcam - ab9106) followed by Alexa-647-conjugated Donkey anti-Rabbit secondary antibody (Jaxson immunoResearch, 711-605-152), following manufacturer's instructions. Nonspecific antibody binding was evident in all brains located mostly around big blood vessels (see Fig. 1c, d). This antibody noise did not affect the quantification as the Myc signal had high signal-to-noise ratio and labeled nuclei were circular in the 3D analysis. For antibodies concentrations, please see "Immunohistochemistry" in "Methods."

**Perfusion, slicing, and mounting**. Mice were given an overdose of Pental and were perfused transcardially with phosphate-buffered saline (PBS) followed by 4% paraformaldehyde (PFA) in PBS. After perfusion, brains were post-fixed in 4% PFA in PBS and then cryoprotected for 24–72 h in 30% sucrose in PBS. To avoid deterioration of the BFP signal, brains that were left in 30% sucrose for >72 h were rinsed in PBS, immersed in O.C.T. gel (Tissue-Tek®), and kept at −70 °C until slicing. Then, 40–60-μm coronal slices of the OB were made using a freezing microtome (Leica SM 2000R) and preserved in PBS. The sections were mounted on slides and cover slipped with mounting media (Vectashield H-1000).

**Scanning and normalization**. From each mouse, we scanned three random slices (both left and right hemispheres) with a confocal microscope (FV1200 Olympus, Japan), using a 20×/0.75 numerical aperture (NA) objective in two channels (differential interference contrast, DIC, and BFP). AbGCs were counted from all slices using random ROI selection within the GCL. To overcome heterogeneities in the image stacks from different slices, the GCL ROIs were normalized with Intensify3D algorithm as described previously[79].

**Automated cell counting**. The normalized images were binarized and cells were counted in 3D using ImageJ. The algorithm output was a binary mask of numerated cells and a statistics table of the cell volumes and surfaces. This output produced >90% accuracy as compared to manual counting. The raw data from ImageJ were analyzed using MATLAB.

**Virus infection quantification**. Regions expressing mRuby were randomly selected within the GCL of each OB using ImageJ. mRuby expressing cells (excitation 561 nm, emission 567 nm) were marked and counted, blind to the BFP channel (excitation 405 nm, emission 430–470 nm). Both channels were then merged, and cells which were found positive in both the BFP channel (indicating abNs) and the mRuby channel (indicating infected neurons) were counted. In the same way, the leakage of the virus was estimated by the number of positive cells in the mRuby channel not containing signal in the BFP channel. All counts were then normalized to a number of cells per volume by the sizes of ROIs and slice thickness. Notably, mRuby expression in the peri-glomerular layer was sparse and nonspecific (Supplementary Fig. 5A–C). The number of DREADD-expressing abGCs was several fold higher than adult-born peri-glomerual neurons, which remained negligible (Supplementary Fig. 5B). In order to quantify the specificity in different layers while avoiding division by zero, we defined specificity index (Supplementary Fig. 5C) as follows: Specificity index = $\frac{\text{#Double labeled cells}+1}{\text{#mRuby labeled cells}+1}$.

**C-Fos evaluation**. Mice were first injected with CNO (5 mg per kg) to silence DREADD-infected adult-born cells, and put back in their home cage. Odor was delivered 1-h post CNO injection, using identical parameters to those of the physiology experiments but directly into their home cage (see odor delivery in "Methods"). Mice were exposed to the odors for 2 h, and then prepared for histology for the assessment of c-Fos expression in both, DREADD-injected and -non-injected bulbs. Two to five slices were then randomly chosen from each hemisphere per each mouse. AbGCs were marked as ROIs using the BFP channel in each slice, blindly to the c-Fos channel (Alexa-647, far red). Channels were then merged, and c-Fos level was assessed as the mean value of pixels within the ROI of each abN. Each slice was then normalized according to its own baseline (assessed by the c-Fos level averaged over the mean value of pixels within the ten ROIs having the lowest values per sample).

**DNA constructs and virus preparation**. *TRE3G-hM4D(Gi)-mRuby2* was constructed using standard molecular cloning methods based on polymerase chain reaction (PCR) and restriction enzymes commercially available from New England Biolabs. In order to make the AAV containing the *TRE3G-hM4D(Gi)-mRuby2* cassette, *hM4D(Gi)* and the *mRuby2* were subcloned using an InFusion kit (Clontech) simultaneously in pAAV-TRE3G-GFP (Tasaka et al.[34]), digested with NcoI and EcoRI. PCR products of *hM4Di* and the *mRuby2* were amplified from pAAV-hSyn-HA-hM4Di (a gift from Bryan Roth; Addgene plasmid #50475; http://n2t.net/addgene:50475; RRID:Addgene_50475) and pAAV-hSyn-mRuby2-GSG-P2A-GCaMP6s (a gift from Tobias Bonhoeffer & Mark Huebener & Tobias Rose; Addgene plasmid #50942; http://n2t.net/addgene:50942; RRID:Addgene_50942)[80], respectively.

**Viral procedure**. We used AAV5.CamKII.GCaMP6f.WPRE.SV40 from Addgene (Cat# 100834-AAV5, $4.1 \times 10^{13}$ genomic copies per ml). AAV1.TRE3g.hM4D(Gi).-mRuby2 virus (DREADD, $1.2 \times 10^{13}$ genomic copies per ml) was produced by the Hebrew University viral vector core. Prior to each injection, both viruses were mixed and injected into the GCL/MCL of the left OB (two injection sites per each bulb, ~300 nl per site, at 300–500-μm depth) using Nanoject (Drummond Scientific). For virus estimation, c-Fos expression evaluation and physiology in anesthetized mice experiments, two small holes were drilled above the OB for the purpose of injection, and immediately sealed. For physiology in awake mice, the bone was removed with a round 2.5-mm diameter biopsy punch (Integra Miltex), and virus injection was followed by a chronic window implantation.

**Immunohistochemistry**. PBS-washed 40-μm thick slices were incubated in the following solutions with gentle agitation: 2 h at room temperature in blocking solution (5% heat inactivated goat serum, 0.4% Triton-X100 in PBS); 3–4 nights at 4 °C in primary antibody 1:1000 mouse anti-Myc (Santa Cruz, Cat #sc-40) or rabbit anti-Fos (1:10,000; Synaptic Systems, Cat #226003) in blocking solution; 2–3 h at room temperature in secondary antibody 1:500 goat anti-mouse-IgG Alexa-647-conjugated or goat anti-rabbit-IgG Alexa-647-conjugated (Jackson ImmunoResearch) in blocking solution; 10 min at room temperature in 2.5 μg/ml of DAPI (4′,6-diamidino-2-phenylindole) (Santa Cruz) in PBS. Each of the above steps was followed by a series of three washes in PBS, 5 min each.

**Odor delivery**. For the odor stimulus presentation, we used a nine-odor air dilution olfactometer (RP Metrix Scalable Olfactometer Module LASOM 2), as described by others[81]. Briefly, the odorants were diluted in mineral oil to 100 ppm. Saturated vapor was obtained by flowing nitrogen gas at flow rates of 100 ml/min through the vial with the liquid odorant. The odor streams were mixed with clean air and adjusted to a constant final flow rate of 900 ml/min. Odors were further diluted tenfold before reaching at a final concentration of 10 ppm to the final valve (via a four-way Teflon valve, NResearch). In between stimuli, 1000 ml/min of a steady stream of filtered air flowed to the odor port continuously. During stimulus delivery, a final valve switched the odor flow to the odor port, and diverted the clean airflow to an exhaust line. Odors were delivered to the mouse nostrils via a custom-made glass mask, at a flow rate of 1 L/min (duration—2 s; interstimulus

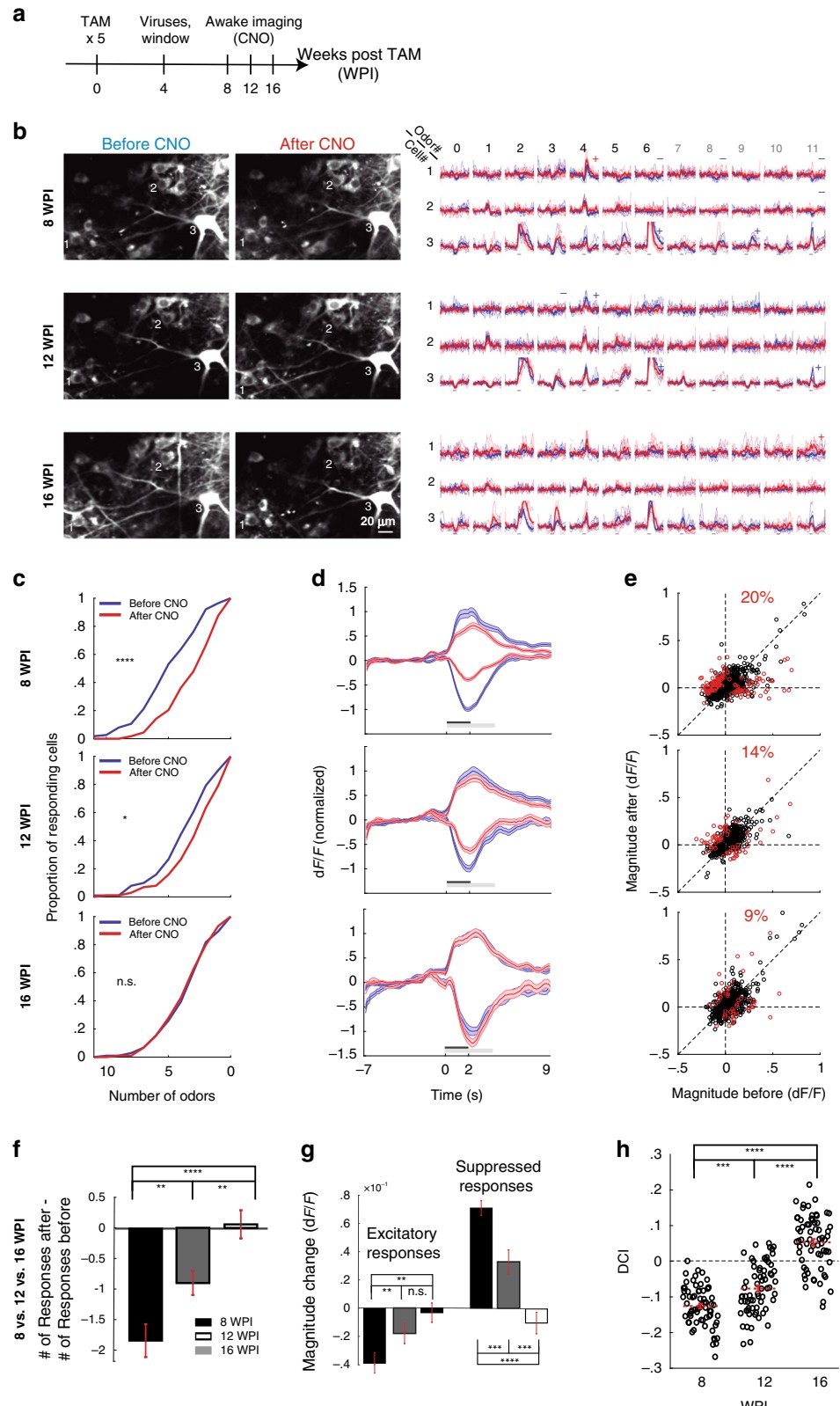

interval—26 s). Odors were continuously removed by air suction. The olfactometer was calibrated using a miniPID (Aurora Scientific).

For pure odors we used a panel of six odorants known to activate different and partially overlapping areas in the dorsal part of the OB (butanal, pentanal, ethyl tiglate, propanal, methyl propionate, ethyl-butyrate, and ethyl acetate; all obtained from Sigma-Aldrich, St. Louis, MO). As six natural odorants we used—male urine, female urine, peanut butter, trimethylthiazoline (TMT), and nest odor. Urine was collected from Nestin X TB males and females (~5 different donors for each urine

mixture) and stored at −20 °C. Twenty-five microliter were placed in the odor vials. Peanut butter was made of 100% peanuts (Better&different, Mishor Edomim, Israel) and 1-g peanut butter was placed in the vials. For predator odor, we used 2 µl of TMT (Contech, Delta, Canada). Nest odor was made of 1-g nest bedding.

**Two-photon calcium imaging**. We performed calcium imaging of the OB using an Ultima two-photon microscope from Prairie Technologies, equipped with a ×16

**Fig. 5 The impact of abGCs on MCs odor coding gradually decreases with age. a** The timeline of the experiment. **b** Left: 2P micrographs of MCs before and after CNO administration, 8, 12, and 16 WPI. Right: calcium transients before (blue) and after (red) CNO injection, from three neurons (marked on the adjacent micrographs). All details are as in Fig. 3b. **c** Cumulative distribution of the proportion of MCs responding to 0–11 odors, across conditions (8 WPI: $n = 113$ cells, $p << 0.0001$; 12 WPI: $n = 101$ cells, $p < 0.05$, 16 WPI: $n = 104$ cells, $p = 0.96$, Kolmogorov–Smirnov tests, two-tailed for 8 WPI, one-tailed for 12, 16 WPI). **d** Traces of the average ± SEM calcium responses to odor stimulation before and after CNO at the three age groups. **e** All responding cell-odor pairs before and after CNO at the three age groups. Red dots are significantly away from the diagonal. The proportions of cell-odor pairs showing significant difference before vs. after were significantly different between the different sessions ($n = 1666$ cell-odor pairs in total, $p << 0.0001$, Chi squared test for equality of proportions). **f** Quantitative analysis of the difference in MCs responsiveness along age ($F_{2,182} = 17.96$, $p << 0.0001$, one-way ANOVA for repeated measures; 8 WPI vs. 12 WPI: $n = 101$ MCs, $p < .01$; 8 WPI vs. 16 WPI: $n = 103$ MCs, $p << 0.0001$; 12 WPI vs. 16 WPI: $n = 92$ MCs, $p < 0.01$, Tukey–Kramer post hoc tests). **g** Quantitative analysis of the difference in response magnitude due to CNO along age (excited responses: $n = 1007$ cell-odor pairs in total, $F_{2,1004} = 5.75$, $p < 0.01$, one-way ANOVA; 8 WPI vs. 12 WPI: $n = 655$, $p < 0.01$; 8 WPI vs. 16 WPI: $n = 707$, $p < 0.01$; 12 WPI vs. 16 WPI: $n = 652$, $p = 0.39$; Suppressed responses: $n = 504$ cell-odor pairs, $F_{2,591} = 38.11$, $p << 0.0001$, one-way ANOVA; 8 WPI vs. 12 WPI, $n = 436$, $p < 0.001$; 8 WPI vs. 16 WPI, $n = 396$, $p << 0.0001$; 12 WPI vs. 16 WPI, $n = 248$, $p < 0.001$; all pairwise comparisons were conducted with Tukey–Kramer post hoc tests). 8 WPI condition: $N = 5$ mice; 12 WPI condition: $N = 4$ mice; 16 WPI condition: $N = 4$ mice, for all comparisons described in this figure. **h** DCIs at the three age groups. ($F_{2,130} = 125$, $p << 0.0001$, one-way ANOVA for repeated measures; 8 WPI vs. 12 WPI; $p < 0.001$, 8 WPI vs. 16 WPI and 12 WPI vs. 16 WPI; $p << 0.0001$, Tukey–Kramer post hoc tests). Statistical tests are two-sided and error bars are SEMs, unless otherwise stated.

water-immersion objective lens (0.8 NA; CF175, Nikon). We delivered two-photon excitation at 920 nm using a DeepSee femtosecond laser (Spectraphysics). Acquisition rate was 7 Hz. Before awake imaging and ~3 weeks after implanting the window, we habituated the mice under the microscope in the head-fixed position. Two-photon microscopy was operated with PrairieView software (version 5.5).

**Data analysis for physiology experiments**. We analyzed all data using Matlab R2018b (Mathworks). Movements were corrected using Moco pluggin (March 2016 release)[82]. Regions of interest corresponding to individual cell bodies were manually drawn and the mean fluorescence of each cell body was extracted by ImageJ at each frame and exported to Matlab for analysis. We calculated relative fluorescence change (dF/F), defining baseline fluorescence (f0) for each cell in each trial as its mean fluorescence measured 5–2 s before odor onset. All traces were smoothed prior to analysis using Matlab's default Smooth function, with a moving average filter with span = 5. The only instance in which non smoothed data were used is for the generation of color bars for Figs. 2i and 3d.

In order to determine the significance of response to odor, a response window was defined as 0–4 s post odor presentation, and local minima/maxima of the mean (over five repetitions) dF/F trace were detected, for cases in which the integral over the response window was negative/positive, respectively. The extremum point was taken together with six adjacent points, three from both sides. According to the type of extremum (minimum or maximum) picked at the response window, a parallel point from a baseline window ranging 6–2 s pre stimulus and a set of six adjacent points were chosen similarly. Effect size for each cell-odor pair was calculated as follows: Effect size $= \frac{\text{Mean}\left(\text{Sample}_{\text{Response}}\right) - \text{Mean}\left(\text{Sample}_{\text{Baseline}}\right)}{\text{Mean}\left(\text{Std}_{\text{response}}, \text{Std}_{\text{Baseline}}\right)}$, where samples are vectors containing seven adjacent data points each, as described above. A response was classified as significant if it had effect size with absolute value bigger than 5, and its magnitude was calculated as the integral over the five repetitions mean trace, 0–5 s post odor delivery (Figs. 2h, k, 3c, g, 5c, e, and Supplementary Fig. 3A). In data recorded from awake animals, where analysis required that a response would be classified as suppressed or excited (Figs. 3b, d, h, 5d, h, and Supplementary Fig. 3A, C), the sign of the integral ($+/-$) determined the type of classification. In order to determine if a response before CNO/saline was significantly different than a response after, a permutation test was conducted on two samples (one before and one after CNO/saline) for each cell-odor pair with at least one significant response (before and/or after), where each sample was composed of five numbers, representing the mean dF/F values averaged over the whole response window for each single trial. Comparisons resulting in $p < 0.05$ were counted as significantly different (Figs. 2e, g, 3b, and 5b, e). Notably, multiple alternative classification methods conducted over the responses have all yielded qualitatively similar results to those reported at the final version of this work. Same is true for different definitions used for responses magnitudes (i.e., extremum vs. integral).

In order to plot separately suppressed and excited mean responses over time in awake experiments (Figs. 3d and 5d), and for further analysis conducted separately on these types of responses, every cell-odor pair classified with a significant suppressed/excited response was taken together with its before/after couple, regardless its classification. This means that in some cases cell-odor pairs were counted twice in this analysis, when a response was detected as significantly excited before and significantly suppressed after (or vice versa). However, these instances constituted <5% of the data set. Since data collected from anesthetized mice did not contain established base of suppressed responses, analysis was conducted only over excited responses. However, the small amount of significantly suppressed $Ca^{2+}$ transients recorded from anesthetized mice, did demonstrate a trend qualitatively similar to that observed in awake mice. This trend had marginal significance, changing with different parameters used for analysis.

For the presentation of responses average over time, normalization was done according to the baseline magnitude as follows: for each type of response class (suppressed/excited), both red and blue average traces were multiplied by a scaling factor determined as abs (1/extremum$_{\text{blue\_trace}}$). For the estimation of empirical tuning curves (Figs. 2j, 3e, f, and Supplementary Fig 3D), all cells-odor pairs were ranked and included, regardless their significance classification. For the estimation of sharpening extent and its significance, a standard deviation (std) was calculated for each cell's tuning curve. Statistical tests were conducted on all cells' stds before vs. after CNO/saline. Sharpening extent was evaluated for further use in the mathematical model over the whole population, as the ratio between mean std before and mean std after manipulation.

Statistical tests were always two-sided, unless stated otherwise. Error bars always represent the standard error of the mean, unless stated otherwise. See Supplementary Table 1 for complete details of all statistics (Supplementary Table 1).

**Estimation of baseline activity from calcium imaging data**. In order to estimate the spontaneous activity in MCs in our data set before vs. after CNO (Supplementary Fig. 2D) we used NES+ mice injected with CNO as our experimental group. As controls we used imaging sessions from awake mice injected with CNO at the absence of DREADD in abNs (Supplementary Fig. 3), and NES+ mice injected with saline (Fig. 3b, bottom). As the experimental group we used NES+ mice injected with CNO at 8 WPI (Fig. 3b, top). We chose a region of interest in the image that served as background (i.e., does not contain any cells). We measured the fluorescence all cell in a given relative to the background ROI, averaged over 60 time points (3–5 s prior to each stimulus). We then subtracted the value before CNO from the value after CNO, yielding either a positive or a negative result for each cell. Positive values indicate an increase and negative values a decrease in baseline activity, respectively.

**Population data analysis**. In order to estimate the ability of a network to discriminate between two given odors, we started by calculating pairwise d primes for each mouse separately (see Fig. 4a, b in which PCA was used for presentation). For two given odors, $p$ and $q$, d primes were calculated according to the formula: $\frac{\text{Distance}(\text{mean}(p), \text{mean}(q))}{\text{Mean}(\text{Inner distance}_p, \text{Inner distance}_q)}$, where $p/q$ is a matrix with five columns (one column for each repetition) and $n$ rows ($n$ = number of cells for each mouse), in which each entry contains the response of a specific cell to a specific odor at a specific trial. $\text{mean}(p)/\text{mean}(q)$ are vectors with $n$ entries representing the mean coordinates (averaged over the five trials) in $n$-dimensional space for odor $p/q$. The difference between these vectors was assessed as a scalar by Euclidean distance, i.e., $\sqrt{\sum_n^{i=1}(\widehat{p}_i - \widehat{q}_i)^2}$. Inner distance$_p$ and inner distance$_q$ were similarly assessed as two scalars, by calculating the Euclidean distance between each one of the five single trials and the average representation, and then averaging over the five resulting measures. Since Euclidean distance measured in $n$-dimensional space increases proportionally to $\sqrt{n}$, it is important to note that both terms, the one in the numerator and the one in the denominator, have the same units. Thus, their ratio cancels this dimensionality dependence and results in a dimensionless quantity, comparable between mice with different number of cells.

We calculated two matrices for each experiment, one for the "before" and one for the "after" CNO/saline condition. Each matrix contained d primes for all 66 possible pairwise comparisons. General matrices for all mice were calculated by a weighted average, where the weight is determined by the number of cells, such that matrices calculated over larger amount of data would have higher impact on the final estimation (Fig. 4c, d).

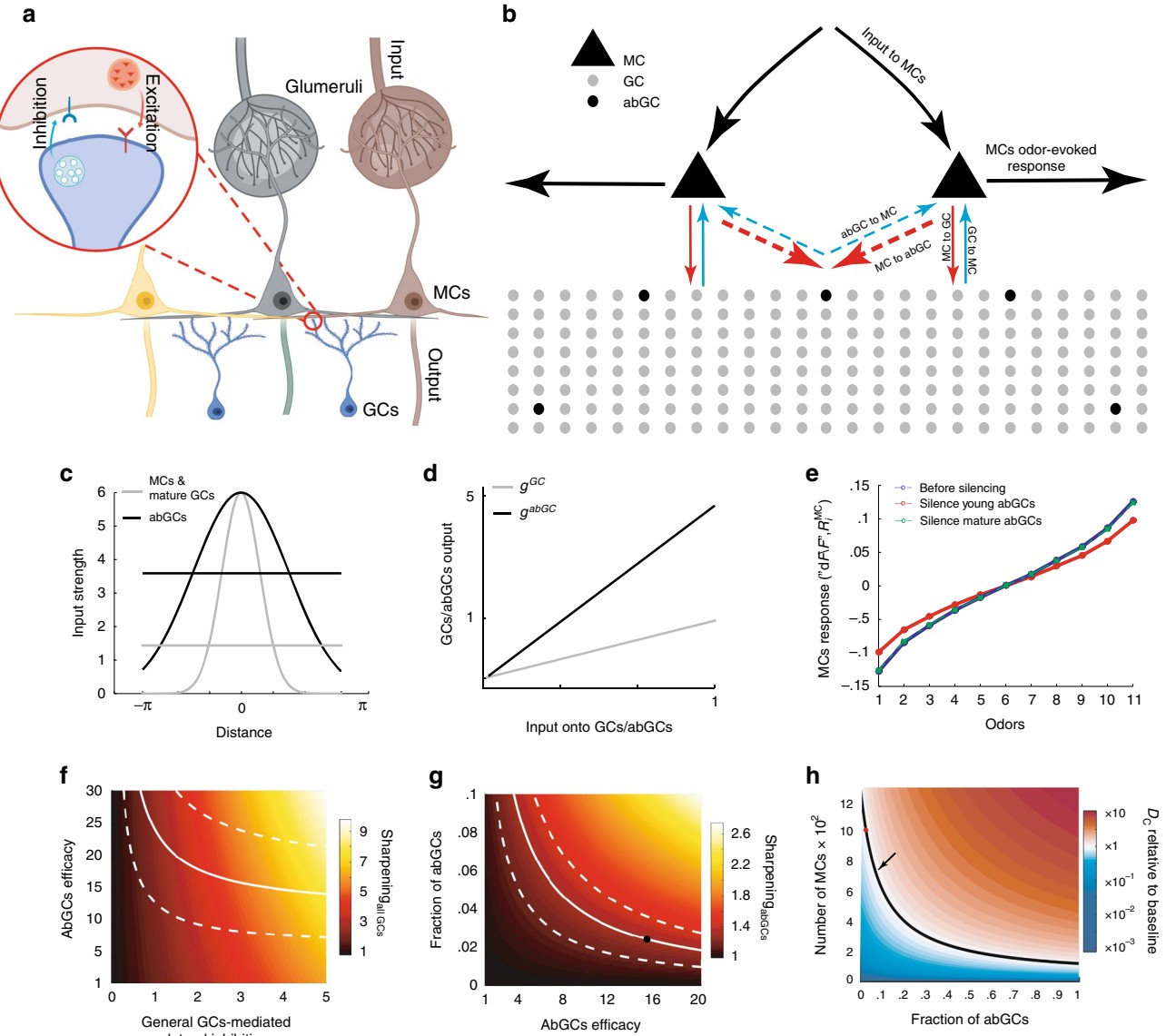

**Fig. 6 A model of the OB suggests a mechanism by which abGCs increase information in MCs. a** Schematics of MCs and GCs/abGCs in the olfactory bulb. **b** A schematic illustration of the model. Blue arrows denote inhibition and red excitation. **c** Connection strength between MCs and GCs/abGCs as a function of spatial proximity. Horizontal lines display the average connectivity based on connectivity width and peak. abGCs have broader, more promiscuous input connectivity than mature GCs. "abGCs input broadness" is the ratio between abGCs and GCs average connectivity. **d** Input–output functions for both GCs and abGCs. "abGCs excitability" is the ratio between abGCs and GCs input–output slope. **e** Silencing young abGCs (2.5% of total GCs) quenches MC responses (red curve, cf. Fig. 3e). Silencing the same amount of mature GCs had no effect (green curve). **f** Sharpening of MC tuning due to entire GC network (Sharpening$_{allGCs}$, i.e., the ratio of MC output tuning to input tuning) as a function of both GC-mediated MC lateral inhibition and abGC efficacy. Efficacy is defined as the product of abGCs excitability (**d**) and their input broadness (**c**). Solid line represents the parameter set estimated by constraining the model to fit our data. Dashed lines—confidence borders (see "Methods"). **g** Sharpening$_{abGCs}$ as measured in our data. Dashed lines— confidence borders (see "Methods"). Black dot represents the parameters used in our model. **h** Number of MCs needed to maintain a fixed level of discrimination between odors (Chernoff distance; see "Methods") as a function of the proportion of young abGCs. Red/blue gradients represent increased/decreased discriminability, respectively. The red dot represents our baseline model network (1000 MCs and 2.5% (2500) abGCs). Black line— fixed value of discrimination in our model network (≈30 $D_c$), for different MCs/abGCs ratios. Arrow—proportion of abGCs (7.5%) that would achieve half the potential reduction in the number of MCs that an all-abGC network would achieve. **a** was created with BioRender.com.

For the quantification of the difference between before vs. after matrices, we calculated DCIs (see Figs. 4e, 5f, and Supplementary Fig. 4C, D) over the averaged matrices as follows: $DCI = \frac{d\ prime\ after - d\ prime\ before}{d\ prime\ after + d\ prime\ before}$. This quantity describes changes in discrimination between two given odors (numerator) with respect to their general discriminability (denominator). Under $H_o$, where no change in discriminability occurs, DCIs should be distributed around 0. Thus, a $t$-test over the mean DCI was used in order to determine significance. We are aware of the fact that the independence assumption required by the $t$-test may be violated here, but we have verified the reproducibility of the reported results between different mice

and conditions, as well as the absence of meaningful discrepancies between odors in our data sets (for impression of the similar trends for all odors, see Supplementary Fig. 4D). Moreover, none of the above described manipulations over the data (i.e., calculating each mouse separately, weighting the average, measuring DCIs instead of raw deltas etc.) qualitatively changed the main results and conclusions.

In addition to the results we refer in the main text body, our analysis revealed that awake mice injected with saline showed a small but significant difference in DCIs. This difference is in the opposite direction to that observed in the same

mice due to the injection of CNO (Fig. 4d, e). This suggests that the effect measured in the experimental condition might be underestimated. We hypothesize that at the absence of a robust perturbation, odor discrimination by MCs may be improved after exposure in awake mice. In accordance with this interpretation, this result was duplicated when awake mice had a similar improvement in DCIs when the mature abGCs were silenced (Fig. 5h, 16 WPI). In anesthetized mice, controls showed no significant differences in DCIs (Supplementary Fig. 4A–C).

**Model definition**. We model the full MC-GC network as a population of linear firing rate neurons with distance-dependent connectivity.

The connectivity from neuron $j$ in population $Y$ to neuron $i$ in population $X$, where $X, Y \in \{\text{MC, GC, abGC}\}$, is given by a distance-dependent Gaussian profile (Fig. 6c)

$$J_{ij}^{XY} = J_{XY} \exp - \frac{(\theta_j - \theta_i)^2}{2\sigma_{XY}^2},\tag{1}$$

where $J_{XY}$ and $\sigma_{XY}$ are the peak strength and breadth of the spatial profile, respectively, for the synapses from population $Y$ to population $X$. For mathematical convenience neurons in each population are arranged uniformly on a ring with $\theta_i \in [0, 2\pi]$. We note that the average connection strength (Fig. 6c, horizontal lines) from population $Y$ to population $X$ is given by:

$$J_{XY}^0 = \frac{J_{XY}\sigma_{XY}}{\sqrt{(2\pi)}}.\tag{2}$$

For simplicity we assume an identical peak connectivity strength between all pairs of populations.

We assume that abGCs do not differ from mature GCs in their output connectivity to MCs

$$\sigma_{\text{MC}\leftarrow\text{abGC}} = \sigma_{\text{MC}\leftarrow\text{GC}},\tag{3}$$

but are more promiscuous in their input connectivity profile:

$$\sigma_{\text{abGC}\leftarrow\text{MC}} = b*\sigma_{\text{GC}\leftarrow\text{MC}},\tag{4}$$

where $b > 1$ is the relative input broadness or promiscuity of abGC input compared to mature GC.

For GCs and abGCs, the input is summed over all MCs and weighted according to the synaptic strength. It is given by:

$$h_i^X = \frac{1}{\#\text{MC}} \sum_{j=1}^{\#\text{MC}} J_{ij}^{X\leftarrow\text{MC}} r_j^{\text{MC}},\tag{5}$$

where $X \in \{\text{GC, abGC}\}$.

The (ab)GC firing rate is a linear function of the input

$$r_i^{\text{GC}} = h_i^{\text{GC}},\tag{6}$$

$$r_i^{\text{abGC}} = g * h_i^{\text{abGC}},\tag{7}$$

where $g > 1$ is the intrinsic excitability of abGCs relative to mature GCs.

The firing rate of MC neuron $i$ in response to odor $k$ is given by:

$$r_i^{\text{MC}}(\text{odor}_k) = I_0 + I_1 z_i^k - \frac{1}{(\#\text{GC}+\#\text{abGC})} \left( \sum_{j=1}^{\#\text{GC}} J_{ij}^{\text{MC}\leftarrow\text{GC}} r_j^{\text{GC}} + \sum_{j=1}^{\#\text{abGC}} J_{ij}^{\text{MC}\leftarrow\text{abGC}} r_j^{\text{abGC}} \right),\tag{8}$$

where the baseline input is $I_0$, and $I_1 z_i^k$ is the odor-specific input. We define $z_i^k$ to be a unit variance, mean-zero, random variable, which we assume to be independent for each MC and each odor. For concreteness we draw these from a std normal distribution, but in fact the following results depend only on $z_i^k$ having finite moments. MC baseline firing rate, $r_i^{\text{MC}}(0)$ is found by setting $z_i^k = 0$.

We define the MC odor-evoked response as the change in firing rate divided by baseline

$$R_i^{\text{MC}}(\text{odor}_k) = \frac{r_i^{\text{MC}}(\text{odor}_k) - r_i^{\text{MC}}(0)}{r_i^{\text{MC}}(0)}.\tag{9}$$

**Odor tuning and model parameters**. In the Supplementary Methods we show that the model can be reduced to three parameters:

(1) The proportion of abGCs: $f$.
(2) The efficacy of abGCs, defined as the product of their increased broadness and excitability (Fig. 6c, d): $g_{\text{eff}} = b * g$.
(3) The average strength of MC-MC lateral inhibition mediated by mature GCs (Fig. 6f): $J_{\text{eff}} = J_{\text{MC}\leftarrow\text{GC}}^0 * J_{\text{GC}\leftarrow\text{MC}}^0$.

These parameters determine the impact of the entire GC population on the tuning properties of MCs.

We define tuning, the sharpness of neuron $i$'s odor tuning, as the std deviation of the odor-evoked response $R_i$ over all odors.

First, we note that without MC $\leftrightarrow$ GC connectivity the extent of tuning of MC cells would be determined directly from the input and given by:

$$\text{Tuning}_{\text{input}} = \frac{I_1}{I_0}.\tag{10}$$

As shown in the Supplementary Methods, the average extent of tuning of MC output under the influence of the MC $\leftrightarrow$ GC connectivity, before silencing abGCs, is

$$\text{Tuning}_{\text{before}} = \frac{I_1}{I_0}(1 + J_{\text{eff}}(1 + (g_{\text{eff}} - 1)f)).\tag{11}$$

As a measure of the GC population's impact on MC tuning, we define $\text{Sharpening}_{\text{allGCs}}$ to be the ratio of extent of tuning of output relative to input (Fig. 6f), and find that this is given in our model by

$$\text{Sharpening}_{\text{allGCs}} = (1 + J_{\text{eff}}(1 + (g_{\text{eff}} - 1)f)).\tag{12}$$

To find the impact of abGCs on MC tuning in our model, we derive the extent of tuning after silencing:

$$\text{Tuning}_{\text{after}} = \frac{I_1}{I_0}(1 + J_{\text{eff}}(1 - f)).\tag{13}$$

Finally, we define $\text{Sharpening}_{\text{abGCs}}$ as the ratio of tuning before and after silencing:

$$\text{Sharpening}_{\text{abGCs}} = \frac{(1 + J_{\text{eff}}(1 + (g_{\text{eff}} - 1)f))}{(1 + J_{\text{eff}}(1 - f))}.\tag{14}$$

We constrain the values of $f$, $J_{\text{eff}}$, and $g_{\text{eff}}$ as follows:

First, we measure tuning empirically for each MC as the std deviation over its odor-evoked responses, both before and after silencing abGCs. $\text{Sharpening}_{\text{abGCs}}$ is then measured as the ratio of population average of $\text{Tuning}_{\text{before}}$ to the population average of $\text{Tuning}_{\text{after}}$. Confidence bounds for this quantity were estimated as the minima and maxima of the following term:

$$\frac{\text{Avg}[\text{Tuning}_{\text{before}}] \pm \text{SEM}[\text{Tuning}_{\text{before}}]}{\text{Avg}[\text{Tuning}_{\text{after}}] \pm \text{SEM}[\text{Tuning}_{\text{after}}]} \#.\tag{15}$$

The measured value of $\text{Sharpening}_{\text{abGCs}}$ constrains a curve through parameter space according to Eq. (14), and the two confidence bounds constrain a region of parameter space around it. For fixed $f$, we constrain a region of the $J_{\text{eff}} - g_{\text{eff}}$ plane (Fig. 6f), and for fixed $J_{\text{eff}}$ we constrain a region of the $g_{\text{eff}} - f$ plane (Fig. 6g).

We emphasize that the abGCs efficacy, $g_{\text{eff}}$, that is necessary to achieve this value for sharpening is a function of $J_{\text{eff}}$, the effective GC-mediated lateral inhibition. For small $J_{\text{eff}}$ the necessary value of $g_{\text{eff}}$ grows without bound, and for the limit of large $J_{\text{eff}}$, $g_{\text{eff}}$ asymptotes to a lower limit of ~12 (Fig. 6f).

Based on existing estimates of overall GC density in the OB, we estimate that the monthly addition of abGCs as labeled in our data accounts for about 2.5% of total GCs. Therefore, we set $f = \frac{\#\text{abGCs}}{\#\text{abGCs}+\#\text{GCs}} = 0.025$, except where otherwise mentioned. After fixing $f$, and fitting $\text{Sharpening}_{\text{abGCs}}$ as described above, there remains one free parameter as elucidated more fully in Supplementary Methods.

For Fig. 6e we set $g = 5$ for the relative gain or intrinsic excitability of abGCs compared to mature GCs and we set the relative input broadness of abGCs to $b = 3$.

Except for Fig. 6f, where connectivity strength is varied, we set the peak strength $J_{XY} = 10$ for all pairs of populations and we set $\sigma_{\text{GC}\leftarrow\text{MC}} = \sigma_{\text{MC}\leftarrow\text{GC}} = 0.5$. This yields $J_{\text{MC}\leftarrow\text{GC}}^0 = J_{\text{GC}\leftarrow\text{MC}}^0 \approx 2$.

**Chernoff distance**. Chernoff distance between two probability distributions, $P$ and $Q$, is defined by

$$D_C = \max_\alpha D_\alpha(P, Q),\tag{16}$$

where

$$D_\alpha = -\log \sum_x P^\alpha(x) Q^{1-\alpha}(x).\tag{17}$$

We find empirically that over all cell-odor pairs the trial-to-trial noise is highly correlated with trial-averaged odor-evoked response. We therefore model the trial-to-trial noise as Poisson, such that a given MC single-trial spike-count in response to odor k is $n_i^{\text{MC}}(\text{odor}_k) \sim Poi(r_i^{\text{MC}}(\text{odor}_k))$.

As we derive in the Supplementary Methods, the expected Chernoff distance between two typical odors for a single MC in our model is

$$D_C \approx \frac{I_1^2}{4I_0}(1 + J_{\text{eff}}(1 + (g_{\text{eff}} - 1)f)).\tag{18}$$

We note that the ratio between Chernoff distance before and after silencing abGCs is approximately the same as the ratio between tuning sharpness before and after, i.e., $\text{sharpening}_{\text{abGCs}}$ calculated above.

By varying the proportion $f$ of abGCs we find the iso-discrimination curve, i.e., the number of MCs needed to maintain a constant value of total Chernoff distance over the population (Fig. 6h). Although our model has one free parameter after fitting to data, the iso-discrimination curve is essentially independent of this degree of freedom (see Supplementary Methods).

**Reporting summary**. Further information on research design is available in the Nature Research Reporting Summary linked to this article.

## Data availability

All data and code that support the findings of this study are available online with detailed instructions at https://github.com/MizrahiTeam/Shani-Narkiss-et-al.-2020..git.

## Code availability

The codes used to analyze the data of current study are available from the corresponding authors with detailed instructions at https://github.com/MizrahiTeam/Shani-Narkiss-et-al.-2020..git.

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

## Acknowledgements

We thank Y. Livneh, E. Lottem, D. Rokni, R. Haddad, and members of the Mizrahi laboratory for commenting on early versions of this manuscript. We thank N. Melamed-Book for her help with confocal microscopy. This work was supported by an ERC consolidators grant to A.M. (#616063), Israeli Science Foundation grant to A.M. (#224/17), German Israeli Foundation grant to A.M. (I-1479-418.13/2018), and the Gatsby Charitable Foundation. Some elements in Figs. 3, 6, and Supplementary Fig. 1 were created with the help of BioRender.com.

## Author contributions

H.S-.N., A.V., and A.M. designed the experiments. G.T. and M.G. prepared viruses. S.T, N.Y., I.M., and H.S-.N. conducted whole brain and slices immunohistochemistry experiments and normalization-based cell counts. H.S-.N. conducted the physiology experiments. H.S-.N. analyzed the data. I.D.L., H.S-.N., and H.S. developed the computational model, H.S-.N. and A.M. wrote the paper.

## Competing interests

The authors declare no competing interests.
