## [Peer Review File · Nature Communications]

Reviewers' Comments:

Reviewer #1:

Remarks to the Author:

The authors present a novel method to target adult born granule cells in the olfactory bulb using tamoxifen induction, permitting the study of the life cycle of adult born granule cells (abGC). The technique is novel and highly specific and will have a broad impact in the field. The paper is not only technical but the authors use the newly developed tool to answer some unresolved questions regarding abGC. The authors show a paradoxical reduction in mitral cell fluorescent responses upon chemogenetic silencing of abGC. The abGC silencing also produced a reduction in the coding capability of mitral cells. Effects of abGC are reduced for older abGC.

The manuscript makes an important contribution that will serve to determine the role played by new neurons being incorporated into existing neural circuits. However, there are still some points that need to be addressed.

Main concerns

1) The authors show that silencing abGC produces an unexpected decrease in the odor evoked responses measured using 2P calcium imaging. This is highly surprising because by silencing a population of inhibitory neurons, one might have concluded that there would be an increase in the odor evoked responses. The reviewer found this extremely puzzling and tried to understand how could silencing an excitatory population would decrease activity, finding the explanations in the results part insufficient. The authors finally suggest in the discussion that the silencing of abGC increase the spontaneous activity of MC (Fo) and because activation of calcium imaging is expressed in terms of dF/F_0 , the increase of the spontaneous activity produces the observed reduction in df/F_0 . The reviewer finds this argument crucial, belonging into the main result. The authors need to show, using electrophysiological measurements that inactivation using CNO increase the spontaneous spiking activity of mitral cells. The argument cannot be in the discussion.

2) One possible explanation for the reduced effect of the silencing on older abGC on the mitral cells could be a reduction in the expression of DREADDs in abGC as the cells age. The authors should determine that the number of abGC at different ages. They should also do immediate early gene experiments to determine that the age of the abGC does not affect their silencing by CNO.

Reviewer #2:

Remarks to the Author:

In this manuscript, titled "How Adult Born Neurons Enhance Circuit Computation", Shani-Narkiss and others describe a new molecular method to label and manipulate adult-born neurons to reveal their role in olfactory processing. The new method involves an expression of inducible Cre (CreER) under the nestin promoter, which labels neuronal progenitor cells. The cre-ER that is induced by tamoxifen in turn drives the expression of tTA, using a line some of the authors developed previously (Tasaka et al., 2018). In this study, the authors demonstrate that the new method can be used to silence abGCs in the OB effectively (tTA-mediated expression of silencer) and use it to analyse how the abGCs at different ages contribute to the sensory tuning of mitral and tufted cells. Curiously, they observe that mitral cell tuning is affected by abGC silencing only when the abGCs are young (about 8 weeks post induction). Further, they demonstrate, using a computational model, that a developmental refinement in the abGC tuning and excitability can explain their experimental observation.

This new method of labeling is efficient, more reliable and convenient compared to previous methods, since all nestin-expressing neurons are the target, and cre can be induced at the time of experimenter's choosing. Since ab Neurons are considered important in other brain areas (e.g., dentate gyrus), and the method described here is also applicable there, too, this study will be relevant

to a general audience.

Most importantly, the crucial contribution that this study provides is a clear, causal evidence as to (1) how abGC contributes to tuning the OB output and (2) what developmental changes in the abGCs may mean for the sensory coding.

Some comments:

(1) The data presented in Figure 1, especially panels D and E, is helpful as it convincingly shows the earliest time window that can be used for manipulation. However, as the silencing takes place at later timepoints, it would be ideal if how many labelled abGCs remain at time points beyond 8PI. This is because abGCs are known to undergo higher turnover rate. If the histology data is not available or difficult to obtain, a simulation using the same model but incorporating reduced number of abGCs (instead of narrowed tuning etc) would be beneficial.

(2) The method is expected to label juxtaglomerular cells and would be an additional target at the virus injection depth concerned (300-500 um). As JG cells are potent regulators of M/TC activity, this is an important parameter to consider. Currently, the only mention is in the methods describing it as "sparse and non-specific, such that it was comparable between control and experiment groups". Here, what are "control" and "experiment" groups? Please clarify so that the readers have a clearer idea about the extent of labelling.

(3) The legend for figure 4 is slightly confusing. For example, for panel (B), it says "All matrices are calculated for the data shown in 'A' only." But later it reads "Colored arrows represent the odors shown in A", and as it implies, there is more data in the matrices than in 'A'. Also, it is unclear what the individual points plotted in (D) correspond to. I guess all odor pairs averaged across mice, but please clarify.

(4) Generally, the model is very well described. However, Figure 6 would benefit from a clearer description/legend. For example, it took some time to find what "Efficacy" was – could it be highlighted better in the methods? Currently it is buried among other details. In addition, it is difficult to understand how the white curves in the panels F and G were obtained. In the methods, it is described as having been derived from experimental data, but it is not clear how the experiments could generate a set of parameters. Please clarify.

(5) Some of the example traces are clipped – e.g., Figure 2E cell 7. Since some of these traces describe the significant change, maybe better to change the scale for some cells. Also, please add scale bars.

(6) Figure 1 - The main text refers to panel F, which should be panel E. (P6,L7; P12,L13) ; Legend: Indicate red circles are the individual mice used in this experiment.

(7) Figure 2, panel I – please add a color map for the t-values

(8) Figure 6, panel C – legend should be corrected from "vertical line" to "horizontal line".

(9) P6,L7 Please change fig. 1F to fig. 1E.

(10) P9,L10-11 Duplicate 'responses'.

(11) P12,L13 Please change fig. 1F to fig. 1E.

(12) P21,L17 NES+ should be NES-.

(13) P28,L10 300 μ L is a very large amount, is the unit correct?

(14) P29,L17 What are the parameters used for the smooth function?

Reviewer #3:

Remarks to the Author:

The manuscript by Mizrahi et al. used long-term in vivo 2-p imaging method to investigate the possible behavioral function of adult-born neurons in olfaction. They found that chemogenetic silencing of new neurons sharpened the mitral cells tuning and improve odor discrimination. Furthermore, they found the effect of new neurons was new neuron age dependent. They finally established a computational model to understand the possible mechanisms. The experiments have been well designed and performed. The model seems to make sense to me, although I was not able to judge and verify the detailed formulas, which are out of my expertise. Overall, it is an important piece of work, and I support publication in general. However, if the authors can revise the followings, the manuscript can be better received.

1) Animal model. This Nestin Cre ERT2 line has been used by several groups, which have been shown to be a useful one. Although the characterization is helpful, given it is a well-characterized line; the authors should try to minimize the introduction and description. It does not carry much new information. The number of marked neurons can be affected by dose of tamoxifen. Therefore, the authors want to state it fairly when compare the number of new neurons with those previous publications. Importantly, once the recombination occurs, most progenitors remain to proliferate for at least several cycles. The authors should provide a fair description of the age of new neurons.

2) The Nes- data has been presented in Figure 2. If there is similar data for other experiments, the authors should include them. If not, the authors should at least discuss about this based on the conclusions in the figure 2.

3) Age dependent experiments. Although whether AAV-delivered transgenes integrate to genome remains being debated, the expression declines after a month or so. It will be necessary for the authors to analyze the expression level of Gi at these ages, which can be assessed by staining. This is necessary to be for sure since it is one of the key findings in this manuscript.

4) The current study has not performed neural circuit study. However, the usage of excitatory and inhibitory responses is rather confusing. Maybe use something like suppression or enhancement can be better so that leave the excitatory and inhibitory terms for describing neuronal physiological properties.

Answer to Reviewers (manuscript # NCOMMS-19-20419360)

We thank the reviewers for their support of the work and constructive comments. We also thank everyone for the patience to resend the revision. Preparing mice for experiments takes many months to prepare, and the pandemic slowed things down considerably. Nevertheless, we have now revised the manuscript and addressed the comments. The new text appears in **bold letters** in the revised manuscript. All answers and changes are detailed below (in blue).

Reviewer #1 (Remarks to the Author):

The authors present a novel method to target adult born granule cells in the olfactory bulb using tamoxifen induction, permitting the study of the life cycle of adult born granule cells (abGC). The technique is novel and highly specific and will have a broad impact in the field. The paper is not only technical but the authors use the newly developed tool to answer some unresolved questions regarding abGC. The authors show a paradoxical reduction in mitral cell fluorescent responses upon chemogenetic silencing of abGC. The abGC silencing also produced a reduction in the coding capability of mitral cells. Effects of abGC are reduced for older abGC.

The manuscript makes an important contribution that will serve to determine the role played by new neurons being incorporated into existing neural circuits. However, there are still some points that need to be addressed.

Main concerns

1) The authors show that silencing abGC produces an unexpected decrease in the odor evoked responses measured using 2P calcium imaging. This is highly surprising because by silencing a population of inhibitory neurons, one might have concluded that there would be an increase in the odor evoked responses. The reviewer found this extremely puzzling and tried to understand how could silencing an excitatory population would decrease activity, finding the explanations in the results part insufficient. The authors finally suggest in the discussion that the silencing of abGC increase the spontaneous activity of MC (Fo) and because activation of calcium imaging is expressed in terms of dF/F_0 , the increase of the spontaneous activity produces the observed reduction in df/F_0 . The reviewer finds this argument crucial, belonging into the main result. The authors need to show, using electrophysiological measurements that inactivation using CNO increase the spontaneous spiking activity of mitral cells. The argument cannot be in the discussion.

Answer: Thanks for this comment. To estimate changes in baseline fluorescence of our data, we added a new analysis to the manuscript. Indeed, we found a significant (though small) increase in basal fluorescence after CNO application. See below and new figure S5D in the revised manuscript. We agree that this data is helpful in interpreting the results.

The graph shows the difference in the pre-odor fluorescence of two groups (experimental and control), before and after CNO. The experimental group showed a significant increase in basal fluorescence after CNO application (significantly higher than 0; $p=0.004$). The control group was not significantly different from 0 ($p=0.74$). The two groups are significantly different ($P=0.02$).

We realize that calcium imaging is not the best method for measurement of absolute firing rates. However, using imaging data has several advantages in the context of this revision; Primarily, our measurements are done on the exact same neurons on which the odor-evoked effects were measured. New experiments with electrophysiology are beyond the scope of this paper for two reasons. First, in such experiments we can yield very few neurons per experiments – i.e. by targeting MCs from the dorsal surface of the OB, and collected data before and after CNO. Second, and more importantly,

please note that others have already shown this result before. Previous work using electrophysiology show clear increase in spontaneous firing rates when silencing abGCs, both *in vitro* (Breton-Provencher et al. 2009; and Mandairon et al. 2018) and *in vivo* (Alonso et al. 2012).

2) One possible explanation for the reduced effect of the silencing on older abGC on the mitral cells could be a reduction in the expression of DREADDs in abGC as the cells age. The authors should determine that the number of abGC at different ages. They should also do immediate early gene experiments to determine that the age of the abGC does not affect their silencing by CNO.

Answer: We agree with this comment. To rule out this possibility, we conducted two new experiments. 1) We quantified the number of DREADD expressing neurons in all ages. See new Fig. S1E, p.11. 2) We performed a new CNO experiment to show that our manipulation is still functional at older ages. Both experiments strengthen the paper. See new Fig. S1D and p.11. See also comment #1 by reviewer #2 and comment #3 by reviewer #3.

Reviewer #2 (Remarks to the Author):

In this manuscript, titled “How Adult Born Neurons Enhance Circuit Computation”, Shani-Narkiss and others describe a new molecular method to label and manipulate adult-born neurons to reveal their role in olfactory processing. The new method involves an expression of inducible Cre (CreER) under the nestin promotor, which labels neuronal progenitor cells. The cre-ER that is induced by tamoxifen in turn drives the expression of tTA, using a line some of the authors developed previously (Tasaka et al., 2018). In this study, the authors demonstrate that the new method can be used to silence abGCs in the OB effectively (tTA-mediated expression of silencer) and use it to analyse how the abGCs at different ages contribute to the sensory tuning of mitral and tufted cells. Curiously, they observe that mitral cell tuning is affected by abGC silencing only when the abGCs are young (about 8 weeks post induction). Further, they demonstrate, using a computational model, that a developmental refinement in the abGC tuning and excitability can explain their experimental observation.

This new method of labeling is efficient, more reliable and convenient compared to previous methods, since all nestin-expressing neurons are the target, and cre can be induced at the time of experimenter’s choosing. Since ab Neurons are considered important in other brain areas (e.g., dentate gyrus), and the method described here is also applicable there, too, this study will be relevant to a general audience.

Most importantly, the crucial contribution that this study provides is a clear, causal evidence as to (1) how abGC contributes to tuning the OB output and (2) what developmental changes in the abGCs may mean for the sensory coding.

Some comments:

(1) The data presented in Figure 1, especially panels D and E, is helpful as it convincingly shows the earliest time window that can be used for manipulation. However, as the silencing takes place at later timepoints, it would be ideal if how many labelled abGCs remain at time points beyond 8PI. This is because abGCs are known to undergo higher turnover rate. If the histology data is not available or difficult to obtain, a simulation using the same model but incorporating reduced number of abGCs (instead of narrowed tuning etc) would be beneficial.

Answer: We agree with this comment. To address it, we conducted a new experiment, quantifying the number of DREAD-infected abGCs at later time points. See new Fig. S1E.

In addition, please note that our model already provides insights and predictions about scenarios of reduced numbers of abGCs (see Fig. 6G). See also comment #2 by reviewer #1 and comment #3 by reviewer #3.

(2) The method is expected to label juxtglomerular cells and would be an additional target at the virus injection depth concerned (300-500 um). As JG cells are potent regulators of M/TC activity, this is an important parameter to consider. Currently, the only mention is in the methods describing it as “sparse and non-specific, such that it was comparable between control and experiment groups”. Here, what are “control” and “experiment” groups? Please clarify so that the readers have a clearer idea about the extent of labelling.

Answer: We now quantify the exact number of DREADD-expressing neurons in the PGL, which were negligible as compared to abGCs (see new Fig. S5A-C). These cells were few and often did not express BFP, suggesting no specificity. This spurious infection was comparable between our experimental and control animals (Nes+ vs. Nes-), as now clarified in the Methods. See p. 28.

(3) *The legend for figure 4 is slightly confusing. For example, for panel (B), it says “All matrices are calculated for the data shown in ‘A’ only.” But later it reads “Colored arrows represent the odors shown in A”, and as it implies, there is more data in the matrices than in ‘A’. Also, it is unclear what the individual points plotted in (D) correspond to. I guess all odor pairs averaged across mice, but please clarify.*

Answer: Both points are now clarified in the text. Indeed, ‘A’ and ‘B’ show example data from 1 mouse, and C-E are the measures from all mice. The DCI values are for all odor pairs averaged across all mice. Clarified on p. 10.

(4) *Generally, the model is very well described. However, Figure 6 would benefit from a clearer description/legend. For example, it took some time to find what “Efficacy” was – could it be highlighted better in the methods? Currently it is buried among other details. In addition, it is difficult to understand how the white curves in the panels F and G were obtained. In the methods, it is described as having been derived from experimental data, but it is not clear how the experiments could generate a set of parameters. Please clarify.*

Answer: Sorry for not being clear on this. This is now further clarified in the text in several places. See p.26 and the additional description regarding the model in the methods section, p.32-36.

(5) *Some of the example traces are clipped – e.g., Figure 2E cell 7. Since some of these traces describe the significant change, maybe better to change the scale for some cells. Also, please add scale bars.*

Answer: We played with this presentation and think its better to leave the scaling the same for all neurons. If the reviewer and editor think otherwise, we will gladly share the visuals and change it. Scale bars are now added.

(6) *Figure 1 - The main text refers to panel F, which should be panel E. (P6,L7; P12,L13) ; Legend: Indicate red circles are the individual mice used in this experiment.*

Answer: Corrected and added.

(7) *Figure 2, panel I – please add a color map for the t-values*

Answer: Added.

(8) *Figure 6, panel C – legend should be corrected from “vertical line” to “horizontal line”.*

Answer: Corrected.

(9) *P6,L7 Please change fig. 1F to fig. 1E.*

Answer: Changed.

(10) *P9,L10-11 Duplicate ‘responses’.*

Answer: Corrected.

(11) *P12,L13 Please change fig. 1F to fig. 1E.*

Answer: Corrected.

(12) *P21,L17 NES+ should be NES-.*

Answer: Sorry for not being clear. Please note that it is indeed NES+ in the absence of tamoxifen.

(13) *P28,L10 300 μ L is a very large amount, is the unit correct?*

Answer: Thanks for spotting this mistake. The correct amount is 300nl.

(14) P29,L17 What are the parameters used for the smooth function?

Answer: Added to the text.

Reviewer #3 (Remarks to the Author):

The manuscript by Mizrahi et al. used long-term in vivo 2-p imaging method to investigate the possible behavioral function of adult-born neurons in olfaction. They found that chemogenetic silencing of new neurons sharpened the mitral cells tuning and improve odor discrimination. Furthermore, they found the effect of new neurons was new neuron age dependent. They finally established a computational model to understand the possible mechanisms. The experiments have been well designed and performed. The model seems to make sense to me, although I was not able to judge and verify the detailed formulas, which are out of my expertise. Overall, it is an important piece of work, and I support publication in general. However, if the authors can revise the followings, the manuscript can be better received.

1) Animal model. This Nestin Cre ERT2 line has been used by several groups, which have been shown to be a useful one. Although the characterization is helpful, given it is a well-characterized line; the authors should try to minimize the introduction and description. It does not carry much new information.

Answer: We left this part particularly because of the reporter, which is new in this context. We think this new combination is worthy of solid characterization. We hope the reviewer agrees that its now balanced.

The number of marked neurons can be affected by dose of tamoxifen. Therefore, the authors want to state it fairly when compare the number of new neurons with those previous publications.

Answer: Yes, our numbers are compared with identical tamoxifen values. This is now clarified in the text.

Importantly, once the recombination occurs, most progenitors remain to proliferate for at lease several cycles. The authors should provide a fair description of the age of new neurons.

Answer: We agree. This is precisely why we always indicated a range rather than a specific age. We are not aware of a better way to describe this.

2) The Nes- data has been presented in Figure 2. If there is similar data for other experiments, the authors should include them. If not, the authors should at least discuss about this based on the conclusions in the figure 2.

Answer: Please note that Figure S3 describes similar control for the awake experiment. However, since we did not find any differences between this and the Saline control, we chose to highlight the Saline controls in the main text. We think this is preferable since it is allows comparison of the same cells/mice under different conditions.

3) Age dependent experiments. Although whether AAV-delivered transgenes integrate to genome remains being debated, the expression declines after a month or so. It will be necessary for the authors to analyze the expression level of Gi at these ages, which can be assessed by staining. This is necessary to be for sure since it is one of the key findings in this manuscript.

Answer: This is a good comment, following which we conducted a new experiment showing that neurons of all ages express DREADD (i.e. mRuby) and that our manipulation is still functional (i.e. c-Fos reduced expression in silenced DREADD+ cells at later time points). See new Fig. S1D,E. See also comment #2 by reviewer #1 and comment #1 by reviewer #2.

4) The current study has not performed neural circuit study. However, the usage of excitatory and inhibitory responses is rather confusing. Maybe use something like suppression or enhancement can

be better so that leave the expiratory and inhibitory terms for describing neuronal physiological properties.

Answer: Fair point. We now changed the terminology throughout the paper to “excited” or “suppressed”.

Reviewers' Comments:

Reviewer #1:

Remarks to the Author:

The manuscript is ready for publication. All my concerns have been successfully addressed.

Reviewer #2:

Remarks to the Author:

The authors have addressed all of my concerns.

Reviewer #3:

Remarks to the Author:

I am satisfied with the responses.